# Viscoelastic mechanics of tidally induced lake drainage in the grounding zone

Hanwen Zhang[1], Richard Katz[1], and Laura A. Stevens[1]

[1]Department of Earth Sciences, University of Oxford, South Parks Road, Oxford, OX1 3AN, UK

**Correspondence:** Hanwen Zhang (hanwen.zhang@earth.ox.ac.uk)

**Abstract.** Drainage of supraglacial lakes to the ice-sheet bed can occur when a hydrofracture propagates downward, driven by the weight of water in the lake. For supraglacial lakes in the grounding zones of Antarctic glaciers, the mechanics of drainage is complicated by their proximity to the grounding line. Recently, a series of supraglacial lake-drainage events through hydrofractures was observed in the Amery Ice Shelf grounding zone, East Antarctica. The lake depth at drainage varied considerably between events, raising questions about the mechanisms that induce hydrofracture even when the lake depth is small. Here we use a modelling approach to investigate the contribution of tidally driven flexure to hydrofracture propagation in the grounding zone. We model the viscoelastic response of a laterally unconfined marine ice sheet to tides, the tidally induced stress, and the contribution of this stress to hydrofracture propagation. A sensitivity analysis explores the dependence of viscoelastic grounding-line dynamics to the material properties of ice and local bedrock bathymetry. We give a model-based criterion that predicts hydrofracture and supraglacial lake drainage as a function of daily maximum tidal amplitude and pre-drainage lake depth in laterally unconfined grounding zones. Although lateral confinement may contribute to the dynamics of lake drainage at Amery, our model predictions agree with observations of hydrofracture events from this grounding zone.

## 1 Introduction

Atmospheric warming is driving increasing meltwater production on ice-sheet and ice-shelf surfaces (Trusel et al., 2015). In the melt season, this meltwater ponds in topographic lows and forms supraglacial lakes. Lakes drain either slowly, through surface drainage channels (Banwell et al., 2019), or rapidly, through hydrofractures (Das et al., 2008). Lake drainages through hydrofractures may impact ice-sheet mass balance in various ways. For grounded ice sheets, hydrofracture efficiently transports surface meltwater to the subglacial hydrological system. This process reduces bed friction and thus modulates ice-flow velocity and flux (Das et al., 2008; Doyle et al., 2013; Tedesco et al., 2013; Stevens et al., 2015; Dunmire et al., 2020). At ice shelves, hydrofractures can initiate or promote rifts. When propagating through ice shelves, rifts can destabilise them by triggering iceberg calving and ice-shelf collapse (Scambos et al., 2000; Glasser and Scambos, 2008; Banwell et al., 2013, 2019; Warner et al., 2021; Lipovsky, 2020), which leads to a loss of buttressing and increased ice-sheet mass loss.

In East Antarctica, satellite imagery suggests that supraglacial lakes often cluster in the grounding zone, particularly at low elevations and bedslopes. Many of these lakes are connected to surface drainage systems or located in regions that are vulnerable to hydrofracturing (Stokes et al., 2019). In the grounding zone, besides lake–water pressure, tensile stress due to

tidal flexure can promote hydrofracturing. Advection of damaged ice produced in the grounding zone could destabilise the downstream ice shelf (Borstad et al., 2012). Thus, it is important to understand how hydrofractures are initiated and promoted in the grounding zone.

Trusel et al. (2022) reported a series of repeated drainage events of a supraglacial lake at the grounding line (GL) of the Amery Ice Shelf, East Antarctica. Interestingly, these drainage events did not occur at a threshold in lake volume, but rather tended to coincide with times of high daily tidal amplitude. These observations raise the question: *how do tides near the GL contribute to the ice-sheet stress field and lake drainage through hydrofracturing*? Trusel et al. (2022) hypothesised that near the GL of the Amery Ice Shelf, the drainage events are promoted by tensile stress due to tidal flexure. A close examination of remotely sensed data (Sect. 4) indicates that the lake studied by Trusel et al. (2022) is in the grounding zone of a laterally confined outlet glacier close to the shear margin of the Amery Ice Shelf. While data analysis by Trusel et al. (2022) shows that lake drainages are correlated with ocean tides, the dynamics at this locale may be complicated by the lateral shear of the upstream outlet glacier (Raymond, 1996; van der Veen and Whillans, 1996) and the lateral confinement of ice flexure (Antropova et al., 2024; Rignot et al., 2024). To give a simpler, more general treatment, we derive a model of tidal-GL migration and associated hydrofracturing of a laterally unconfined marine ice sheet. This approach enables a detailed study of viscoelastic, tidal-GL dynamics and their sensitivity to ice material properties and local bed geometry.

The GL is the internal boundary between the grounded ice sheet and floating ice shelf. Variations in its position affect the ice flux from inland to the sea. In Antarctica, the GL response to diurnal ocean tides has been documented by various observations. On the Rutford Ice Stream, West Antarctica, ice flow is modulated by semi-diurnal tides, with tidal effects on the ice-flow rate propagating tens of kilometres upstream (Gudmundsson, 2006; Murray et al., 2007; Minchew et al., 2017). At the Amery Ice Shelf, kilometre-scale tidal-GL migration with seawater intrusion has been observed from differential radar interferometry (Chen et al., 2023). The observed grounding zone is much larger than predicted from hydrostatic equilibrium, raising questions about whether the observed tidal flexure within the grounding zone is associated with stresses that contribute to hydrofracturing.

Ice-shelf flexure at the GL can be modelled using thin-plate theory with an elastic (Vaughan, 1995; Sayag and Worster, 2011; Wagner et al., 2016; Warburton et al., 2020) or a viscoelastic constitutive relationship (Reeh et al., 2003; Gudmundsson, 2007). In these models, the GL is treated as a peeling front or as the clamped end of the ice shelf. Thin-plate models capture the large-scale flexure of ice shelves, and neglect the membrane stress that is associated with lateral extension. Moreover, various studies have used a vertically integrated theory with a viscous constitutive law to investigate the dependence of steady-state GL position on ice thickness, sliding laws, buttressing effects, and bed topography (Schoof, 2007a, b; Katz and Worster, 2010; Schoof, 2012; Tsai et al., 2015; Pegler, 2018; Haseloff and Sergienko, 2022; Sergienko and Haseloff, 2023). Besides depth-integrated models, full-Stokes models have also been used to study the migration of GLs on both longer timescales (Nowicki and Wingham, 2008; Durand et al., 2009; Favier et al., 2012; Gudmundsson et al., 2012; Cheng et al., 2020) and tidal timescales (Gudmundsson, 2011; Rosier et al., 2014, 2015; Rosier and Gudmundsson, 2020). In these models, the ice-sheet–bed contact problem has been incorporated as boundary conditions. A more recent development by Stubblefield et al. (2021) and de Diego et al. (2022) has incorporated the contact boundary conditions into variational inequalities. This formulation

enabled the representation of the contact condition within a variational framework that was implemented in the Finite Element method. In this study, we adopt the framework by Stubblefield et al. (2021) to study the tidal effects on GL dynamics of an idealised viscoelastic Ice Shelf.

For a Maxwell viscoelastic ice shelf subject to external tidal forcings, the shelf initially responds elastically, followed by viscous creep. The Maxwell time represents the characteristic timescale over which the ice transitions from behaving elastically to viscously. The Maxwell time of ice ranges from hours to weeks, depending on the local ice properties and stress state. Since the semi-diurnal tidal period (approximately 12 hours) lies within this range, we use a viscoelastic constitutive relationship to model the tidal flexure. The model encompasses both the elastic limit and viscous limit of ice dynamics, making it applicable to a spectrum of glaciers with varying material properties and tidal-forcing frequencies. In particular, we extend the framework for a marine ice sheet with viscous ice flow by Stubblefield et al. (2021) to an upper-convected Maxwell (UCM) model (e.g., Snoeijer et al., 2020) to capture the viscoelastic stress and GL migration induced by the large deformation associated with tides. The upper convected time derivative provides an objective measure of the rate of stress change within a fluid material parcel, meaning that the stress response is independent of the observer's frame of reference.

We use this framework to predict the tensile stress at the GL during daily maximum tidal amplitudes, and conduct an analysis of the sensitivity of this tensile stress to ice rheology and bed topography. Then, using Linear Elastic Fracture Mechanics (LEFM) analysis, we estimate the contributions from tidal stress and lake-water supply to quasi-static hydrofracturing. This enables a model-based criterion in terms of tidal amplitude and lake-water depth for tidally-induced supraglacial lake drainage. The results indicate that tidal flexure could contribute to supraglacial lake drainage through hydrofracturing in a laterally unconfined ice shelf. We apply the model-based criterion to the lake that was studied at the Amery Ice Shelf (Trusel et al., 2022) and compare our model predictions with observations. We make this comparison with the caveat that this lake may be influenced by lateral shear stress that is not included in our model.

The paper is organised as follows. In Sect. 2, we introduce the viscoelastic marine ice-sheet model and the corresponding numerical implementation. Sect. 3 demonstrates the viscoelastic tidal response of a marine ice sheet, followed by an analysis of the model's sensitivity to *(i)* the Deborah number, a ratio of ice Maxwell time to tidal forcing period, and *(ii)* bedslope angle. In Sect. 4, we establish a model-based drainage criterion and compare it with lake-drainage observations from the Amery grounding zone. Finally, we discuss the limitations of this model, and explain how the model could be improved to make predictions of lake drainage in laterally confined glaciers.

## 2  Method

We adopt the viscous, marine ice-sheet flow-line model by Stubblefield et al. (2021) and incorporate a viscoelastic constitutive law. In this section, we introduce the model set-up, including the governing equations and boundary conditions for a viscoelastic marine ice sheet, and how we solve these numerically.

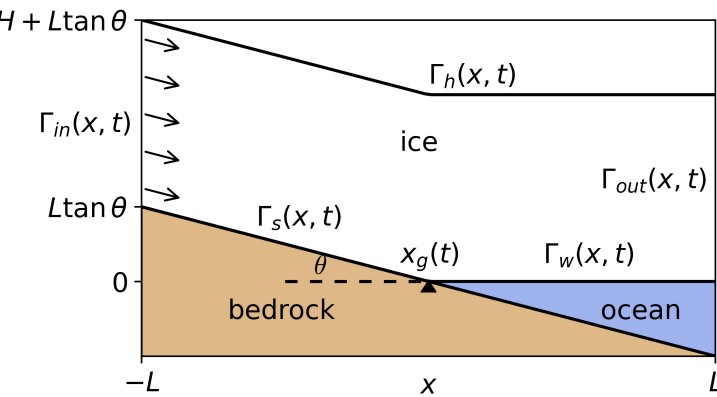

**Figure 1.** Schematic showing the model domain of a marine ice sheet system.

## 2.1 Model domain

Fig. 1 shows a schematic of the computational domain. We consider a segment of marine ice sheet with length $2L$ and thickness $H(x,t)$ in a Cartesian coordinate system with position vector $\boldsymbol{x} = (x, z)$, where $z$ increases upward. The inflow and outflow

boundaries are denoted $\Gamma_{in}$ and $\Gamma_{out}$. The top surface is denoted $\Gamma_h$. The bottom is divided into two parts, according to whether the ice is in contact with the bedrock or the ocean. The ice–bedrock interface $\Gamma_s$ is where ice is in contact with the bedrock at a height $b(x)$. As a simplification, we assume that the bedrock has a uniform slope $\theta$. The ice–ocean interface $\Gamma_w$ is where ice is detached from the bedrock. The two boundaries $\Gamma_s$ and $\Gamma_w$ meet at the GL whose horizontal position, denoted $x_g(t)$, migrates with time $t$. The origin of the coordinate system is set at the middle of the domain on the ice–bedrock interface (at the position

of the GL shown in the schematic).

Following Stubblefield et al. (2021), we first construct the mesh with piecewise linear bottom profile $s(x)$ and surface profile $h(x)$

$$s(x) = \max(b(x), 0),$$
$$h(x) = s(x) + H,$$
(1)

which are shown in Fig. 1. We evolve this initial profile with no tides until the ice-flow geometry (i.e., $s(x)$, $h(x)$ and GL $x_g$)

reach a steady state. In practice, the ice flow reaches the steady state when the GL attains its steady position, defined as when its migration can not be resolved with the 25-m grid size. This provides a steady profile of the marine ice sheet for use as an initial condition for simulations with tides.

## 2.2 Governing equations

The governing equations for momentum and mass conservation are

$$\nabla \cdot \boldsymbol{\sigma} + \rho_i \boldsymbol{g} = \boldsymbol{0}, \tag{2}$$

$$\nabla \cdot \boldsymbol{u} = 0, \tag{3}$$

where $\boldsymbol{\sigma}$ is the total Cauchy stress tensor, $\rho_i$ is ice density, $\boldsymbol{g}$ is gravity, and $\boldsymbol{u}$ is the ice velocity field. The stress $\boldsymbol{\sigma}$ can be decomposed as an isotropic part and deviatoric part, $\boldsymbol{\sigma} = -p\boldsymbol{I} + \boldsymbol{\tau}$, where $p$ and $\boldsymbol{\tau}$ represent the pressure and deviatoric stress, respectively. Here $\boldsymbol{I}$ is the unit tensor.

To model viscoelasticity, we adopt the upper-convected Maxwell formulation for the deviatoric stress $\boldsymbol{\tau}$. The constitutive relationship is

$$\boldsymbol{\tau} + \lambda \overset{\nabla}{\boldsymbol{\tau}} = 2\eta \dot{\boldsymbol{\varepsilon}}, \tag{4}$$

where the Maxwell time, $\lambda = \eta/\mu$, is the ratio of ice viscosity $\eta$ to shear modulus $\mu$. The strain rate is denoted $\dot{\boldsymbol{\varepsilon}}$. The upper-convected time derivative (Oldroyd rate) $\overset{\nabla}{\boldsymbol{\tau}}$ measures the temporal variation of $\boldsymbol{\tau}$ including the effect of rigid body rotation,

$$\overset{\nabla}{\boldsymbol{\tau}} = \partial_t \boldsymbol{\tau} + \boldsymbol{u} \cdot \nabla \boldsymbol{\tau} - (\nabla \boldsymbol{u})^T \cdot \boldsymbol{\tau} - \boldsymbol{\tau} \cdot \nabla \boldsymbol{u}, \tag{5}$$

where $(\cdot)^T$ represents tensor transpose.

We assume a constant shear modulus $\mu$ and non-Newtonian viscosity $\eta$ that is governed by Glen's flow law with regularisation

$$\eta = \frac{1}{2} B \left( |\dot{\boldsymbol{\varepsilon}}|^2 + \delta_\nu \right)^{-(n-1)/2n}, \tag{6}$$

where $B = 2^{(n-1)/2n} A_0^{-1/n}$ is determined by the two flow law parameters $A_0$ and $n$, and $|\dot{\boldsymbol{\varepsilon}}| = \sqrt{\dot{\boldsymbol{\varepsilon}} : \dot{\boldsymbol{\varepsilon}}}$ is the Frobenius norm of the strain rate. The regularisation with the numerical parameter $\delta_\nu$ is used to prevent infinite viscosity at vanishing strain rate (Jouvet and Rappaz, 2011; Helanow and Ahlkrona, 2018; Stubblefield et al., 2021). The value of $\delta_\nu$ sets an upper limit on the viscosity and, therefore, also on the Maxwell time. In our reference parameter set, used below, $\eta \leq 1.3 \times 10^{14}$ Pa s and $\lambda \leq 120$ hr. For $\delta_\nu = 0$, Eq. 6 reduces to the classical form of Glen's flow law.

## 2.3 Boundary conditions

Neglecting atmospheric pressure and other surface loading, the top boundary is assumed to be stress-free. Its elevation $h$ is governed by the kinematic condition

$$\frac{\partial h}{\partial t}(x,t) = \left[ \left( \frac{\partial h}{\partial x} \right)^2 + 1 \right]^{1/2} \boldsymbol{u} \cdot \boldsymbol{n} \qquad \text{on } \Gamma_h, \tag{7}$$

where $\boldsymbol{n}$ is the outward-pointing unit normal vector and $\Gamma_h$ is the top boundary.

On the inflow boundary $\Gamma_{in}$, we impose a uniform horizontal inflow rate $u_0$ and zero shear stress

$$
\begin{cases}
\boldsymbol{u} \cdot \boldsymbol{n} = u_0, \\
\boldsymbol{t} \cdot \boldsymbol{\sigma} \cdot \boldsymbol{n} = 0,
\end{cases}
\quad \text{on } \Gamma_{in},
\tag{8}
$$

where $\boldsymbol{t}$ is the tangential unit vector. The inflow velocity $u_0$ is set to be the satellite-derived surface velocity, $18 \, \mathrm{m \, y^{-1}}$ (Rignot et al., 2016a). On the outflow boundary, we impose the ice-overburden pressure

$$
\boldsymbol{\sigma} \cdot \boldsymbol{n} = -\rho_i g \left(h - z\right) \boldsymbol{n}, \qquad \text{on } \Gamma_{out},
\tag{9}
$$

which means that at the downstream boundary $\Gamma_{out}$, the ice shelf floats at hydrostatic equilibrium, without bending stress.

Similar to Eq. 7, the bottom profile $s\left(x,t\right)$ is governed by the kinematic equation

$$
\frac{\partial s}{\partial t}(x,t) = - \left[ \left( \frac{\partial s}{\partial x} \right)^2 + 1 \right]^{1/2} \boldsymbol{u} \cdot \boldsymbol{n}, \qquad \text{on } \Gamma_w,
\tag{10}
$$

where $\Gamma_w$ is the ice–ocean interface. The stress on the bottom boundary depends on the local contact condition. To introduce the boundary conditions related to the contact problem, we consider hydrostatic water pressure $p_w$ on the ice–ocean interface, defined as

$$
p_w = \rho_w g \left[ h_w\left(t\right) - s^* \right], \qquad \text{on } \Gamma_w,
\tag{11}
$$

where $\rho_w$ is water density, $g$ is gravitational acceleration, $h_w$ is the sea level and $s^*$ is the approximated bottom boundary that will be introduced in subsection 2.4. The atmospheric pressure is neglected at the ice–ocean interface, as it is generally small compared to the hydrostatic pressure.

The sea level $h_w$ is a superposition of a steady state $h_0$ and a sinusoidal function of time, representing ocean tides with amplitude $A$ and frequency $f$,

$$
h_w\left(t\right) = h_0 + A \sin\left(2\pi f t\right).
\tag{12}
$$

On the ice–ocean interface, the hydrostatic pressure $p_w$ is imposed as the traction

$$
\boldsymbol{\sigma} \cdot \boldsymbol{n} = -p_w \boldsymbol{n}, \qquad \text{on } \Gamma_w.
\tag{13}
$$

On the ice–bedrock interface, ice can be either attached or detached from the bed. In the normal direction, the contact condition is established by the following boundary conditions

$$
\begin{cases}
\sigma_n \geqslant p_w, \\
u_n \leq 0, \qquad \text{on } \Gamma_s, \\
\left( \sigma_n - p_w \right) u_n = 0,
\end{cases}
\tag{14}
$$

where $\sigma_n$ is the normal component of traction. Here the water pressure $p_w$ follows Eq. 11. When $u_n = 0$, ice is attached to the bed, $\sigma_n \geq p_w$. When $u_n < 0$, ice is detached from the bed, thus $\sigma_n = p_w$. The impenetrability condition is implemented using a penalty term shown in subsection 2.4, originally proposed by Stubblefield et al. (2021).

In the tangential direction, ice sliding is resisted by friction that is governed by a Weertman-type sliding law (Weertman, 1957)

$$\boldsymbol{t} \cdot \boldsymbol{\sigma} \cdot \boldsymbol{n} = -C \left[ (\boldsymbol{u} \cdot \boldsymbol{t})^2 + \delta_s \right]^{-\frac{n-1}{2n}} \boldsymbol{u} \cdot \boldsymbol{t} \qquad \text{on } \Gamma_s, \tag{15}$$

where $C$ is the friction coefficient, $\delta_s$ is a numerical factor preventing singularity, and $n$ is the exponent in Glen's flow law Eq. 6. In the computation, we choose $C$ such that the surface velocity at the GL matches the inflow speed $u_0$. This choice gives a relatively low surface velocity gradient, which agrees with satellite observations at the lake region (Rignot et al., 2016a).

## 2.4 Numerical Implementation

When implementing the hydrostatic water pressure on the ice–ocean interface, for numerical stability, rather than the bottom elevation from the previous time step, $s^*$ is an approximation to the current step elevation (Durand et al., 2009; Stubblefield et al., 2021), defined as

$$s_*(x,t) = s(x, t - \Delta t) - u_n(x, s, t)\Delta t, \tag{16}$$

where $\Delta t$ is the numerical time step, $s(x, t - \Delta t)$ is the bottom profile at the previous time step, and $u_n$ is the normal velocity on the bottom boundary ($u_n > 0$ accounts for downward motion).

In the variational formulation, on the ice–bed interface, the contact condition (Eq. 14) is accounted for by imposing the hydrostatic pressure (Eq. 11) to basal ice, along with a line integral as a penalty term in the variational formulation

$$\frac{1}{\varepsilon} \int_{\Gamma_s} \frac{1}{2} \left( \boldsymbol{u} \cdot \boldsymbol{n} + |\boldsymbol{u} \cdot \boldsymbol{n}| \right) \boldsymbol{v} \cdot \boldsymbol{n} \, \mathrm{d}s, \tag{17}$$

where $\varepsilon$ is the penalty parameter and $\boldsymbol{v}$ is the test function corresponding to the velocity field $\boldsymbol{u}$. The penalty term (Eq. 17) becomes non-zero only when $u_n > 0$, and thus penalises penetration. For the viscous contact problem, when $\varepsilon \to 0$ the solution to the variational formulation weakly converges to the solution governed by the contact condition (Eq. 14) (Kikuchi and Oden, 1988). For the viscoelastic case, although the variational formulation cannot be directly cast as a minimization problem for $\varepsilon \to 0$, this limit is still a good approximation to the solution governed by the contact condition.

The variational formulation is implemented using the finite-element library FEniCS (Logg and Wells, 2010; Logg et al., 2012; Langtangen and Logg, 2017). A mixed finite element is used to solve for a combined field $(\boldsymbol{u}, p, \boldsymbol{\tau})$. We use triangular elements in which the pressure varies linearly and the velocity and deviatoric stress vary quadratically. We report convergence tests showing the results are mesh-independent (Appendix A). Meanwhile, in the limit of no elastic deformation ($\mu \to \infty$), the model results converge to the viscous solutions by Stubblefield et al. (2021) (Sect. 3). For further details about the variational formulation and its numerical implementation, the reader is referred to Stubblefield et al. (2021).

**Table 1.** Parameters used in numerical model and their reference values.

| Physical property | Notation | Value |
|---|---|---|
| Density of water | $\rho_w$ | $1027$ kg m$^{-3}$ |
| Density of ice | $\rho_i$ | $917$ kg m$^{-3}$ |
| Length of the domain | $L$ | $20$ km |
| Ice thickness | $H$ | $500$ m |
| Bedslope angle | $\theta$ | $0.02$ |
| Glen's Law exponent | $n$ | $3$ |
| Viscosity coefficient | $A_0$ | $3.1689 \times 10^{-24}$ Pa$^{-n}$ s$^{-1}$ |
| Characteristic (inflow) velocity | $u_0$ | $9$ m y$^{-1}$ |
| Friction coefficient | $C$ | $1.2 \times 10^7$ Pa$^{1/n}$ m$^{-1}$ |
| Shear modulus | $\mu$ | $0.30 \times 10^9$ Pa |
| Viscosity regularisation parameter | $\delta_\nu$ | $10^{-18}$ s$^{-2}$ |
| Friction regularisation parameter | $\delta_s$ | $10^{-15}$ m$^2$s$^{-2}$ |
| Penalty parameter | $\varepsilon$ | $10^{-13}$ |
| Tidal amplitude | $A$ | $1$ m |
| Ice fracture toughness | $K_C$ | $10^5$ Pa m$^{1/2}$ |

## 3 Results

We first present a reference case representing the tidal response of an idealised marine ice sheet without lateral confinement. We consider a 20-km long section of a 500-m thick ice sheet, sliding on bedrock with constant bedslope angle $\theta = 0.02$ (Fig. 2). Initially, the grounded ice sheet and floating ice shelf each cover 10 km of the domain. In the model of elasticity, we use Young's modulus $E = 0.88$ GPa and Poisson's ratio $\nu = 0.41$, as suggested by Vaughan (1995) for tidal flexure problems. A list of parameters and their references values are provided in Table 1. The tidal amplitude $A$ is chosen to be 1 m. Except for the lateral confinement, the idealised model is designed to mimic the real grounding zone reported in Trusel et al. (2022). Sect. 4 provides a detailed analysis of the remotely-sensed ice velocity and stress within that specific grounding zone.

### 3.1 Tidally-induced grounding line migration and stress

As shown in Eq. 1, we first construct the mesh with a piecewise linear bottom profile and evolve this initial profile with no tides until the ice-flow geometry (i.e., $s(x)$, $h(x)$ and GL $x_g$) reaches a steady state. The steady profile is then used as an initial condition for simulations with tides.

We find tidally modulated GL migration and corresponding changes in stress. The GL position $x_g$ is shown in Fig. 2e. Whereas Stubblefield et al. (2021) find double GLs at low tides with a relatively small bedslope angle $\theta = 2.5 \times 10^{-4}$ (Stub-

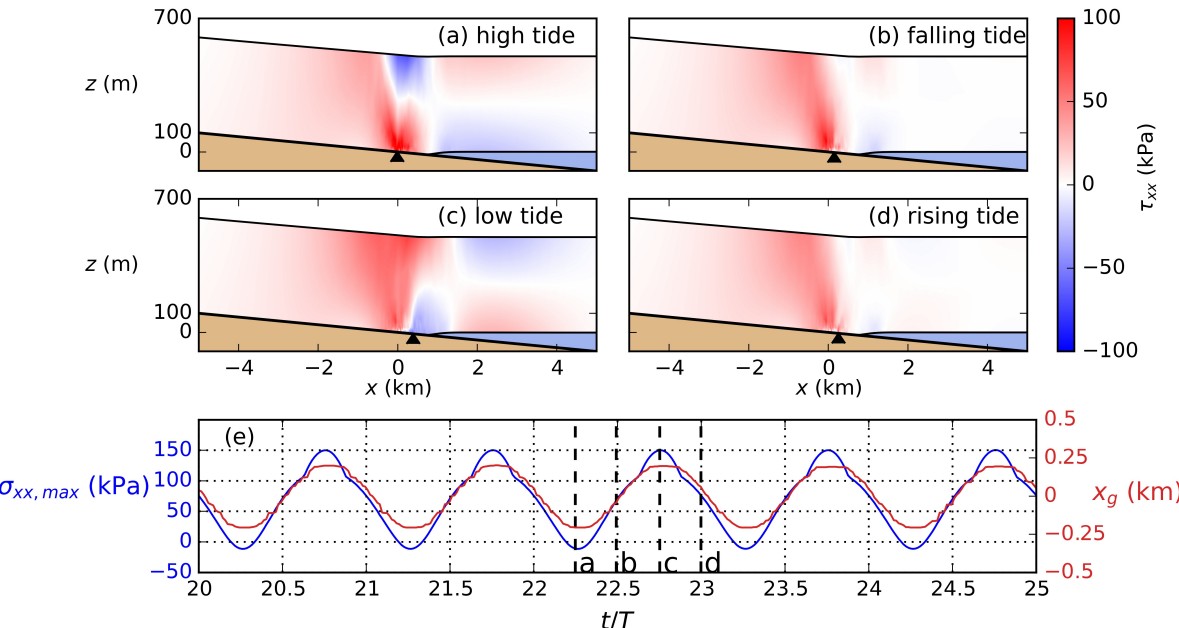

**Figure 2.** Tidal response of a marine ice sheet at different tidal phases. **(a)–(d)** Deviatoric tensile stress $\tau_{xx}$ in one tidal period, with red indicating tensile stress and blue indicating compressive stress. The black triangle marks the position of the GL. **(e)** The maximum tensile stress $\sigma_{xx,max}$ (blue) on the top boundary within the lake region ($\bar{x}_g - 0.5$ km $\leq x \leq \bar{x}_g + 0.5$ km) and the GL position $x_g$ (red) versus time (scaled by the tidal period $T$) with positive values representing downstream migration. Vertical dashed lines show the time of panels (a)-(d).

blefield et al., 2021), in our model we find only a single GL migrating in phase with the tides. This migration results in a 600-m wide grounding zone, which is larger than estimated from hydrostatic equilibrium assuming a uniform ice thickness

($2A/\theta = 100$ m).

To demonstrate tidal flexure, we plot the deviatoric stress component $\tau_{xx}$ at four tidal phases (Fig. 2a–d). The ice undergoes upward and downward flexure at high and low tides, respectively. At high tide (Fig. 2a), the stress is concentrated close to the GL, with compression near the top and tension near the bottom, closely downstream of the GL. This resembles the stress pattern of a thin-plate (Timoshenko, 1955), indicating a region where the ice vertical velocity transitions from the ice-sheet flow

to the ice-shelf oscillation with tides. Further downstream at the floating shelf, the stress is predominantly cryostatic without bending. At low tide (Fig. 2c), the tensile stress dominates the ice-sheet top surface near the GL. The region experiencing tensile stress is larger and located further upstream. At rising tides (Fig. 2b) and falling tides (Fig. 2d), $\tau_{xx}$ is tensile at the GL, but the magnitude is smaller than $\tau_{xx}$ at low tide.

The full horizontal tensile stress $\sigma_{xx}$ is considered for hydrofracturing at the lake. Assuming that the lake covers the ice-

sheet surface within $|x - \bar{x}_g| \leq 0.5$ km, where $\bar{x}_g$ is the time-averaged GL position in a tidal period, we calculate the maximum $\sigma_{xx}$ on the ice-sheet surface within the lake region for any given time, which is denoted $\sigma_{xx,max}$. The temporal variation of

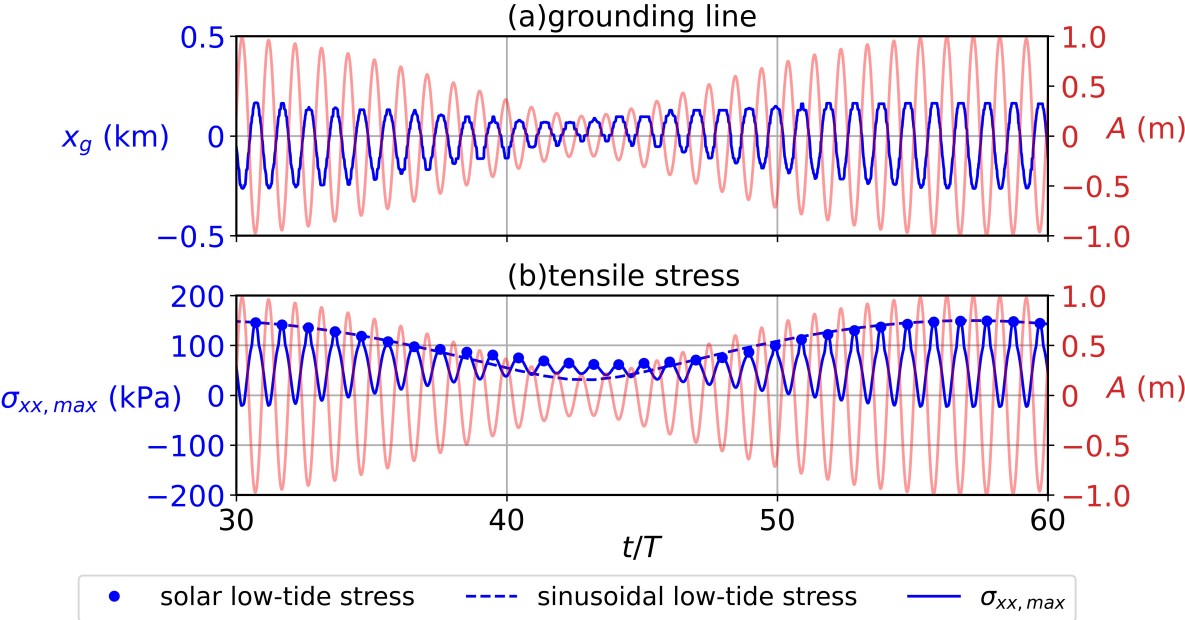

**Figure 3. (a)** Modulated tidal amplitude (red) and corresponding GL migration (blue). The horizontal axis is time scaled by the tidal period. **(b)** (blue) Maximum deviatoric tensile stress $\sigma_{xx,max}$ on the ice-sheet surface within the lake region with modulated tidal amplitude. The dots denote the low-tide stress in one tidal period. The dashed blue line is the estimated low-tide stress calculated from the modulated tidal amplitude using the $\sigma-A$ relationship from sinusoidal semi-diurnal tides in subsection 3.1.

$\sigma_{xx,max}$ is shown in Fig. 2e. In each tidal period, $\sigma_{xx,max}$ reaches its peak at low tide, corresponding to the downward flexural stress in Fig. 2c.

The reference case gives the tidal stress at tidal amplitude $A = 1$ m. We further consider cases with a series of tidal amplitudes from 0 to 1 m, and thus obtain a stress–amplitude relationship for sinusoidal semi-diurnal tides, which we refer to as the "$\sigma-A$ relationship" from this point forward. Specifically, this is the relationship between $\sigma_{xx,max}$ at low tides and the tidal amplitude $A$, assuming that the sea-level variation due to tides follows a monochromatic sinusoid over time. However, with solar tides, tidal amplitude is modulated in a two-week cycle. Given viscoelastic rheology has history-dependence, such an amplitude modulation might complicate the $\sigma-A$ relationship from monochromatic tides.

To explore the $\sigma-A$ relationship with solar tides, we replace the sine function in Eq. 12 with a sine function whose amplitude is modulated over a 14-day period, with sea-level variation shown in Fig. 3. Applying this forcing to the reference case, the GL migrates in phase with tides (Fig. 3a). In each tidal period the low-tide $\sigma_{xx,max}$ tracks the $\sigma-A$ relationship for sinusoidal tides, with slight discrepancies observed at small tidal amplitudes (Fig. 3b), indicating that solar tidal amplitude modulation does not change the $\sigma-A$ relationship. Therefore, daily maximum tidal amplitude proves to be a good metric to estimate the daily maximum tidal stress that contributes to hydrofracturing.

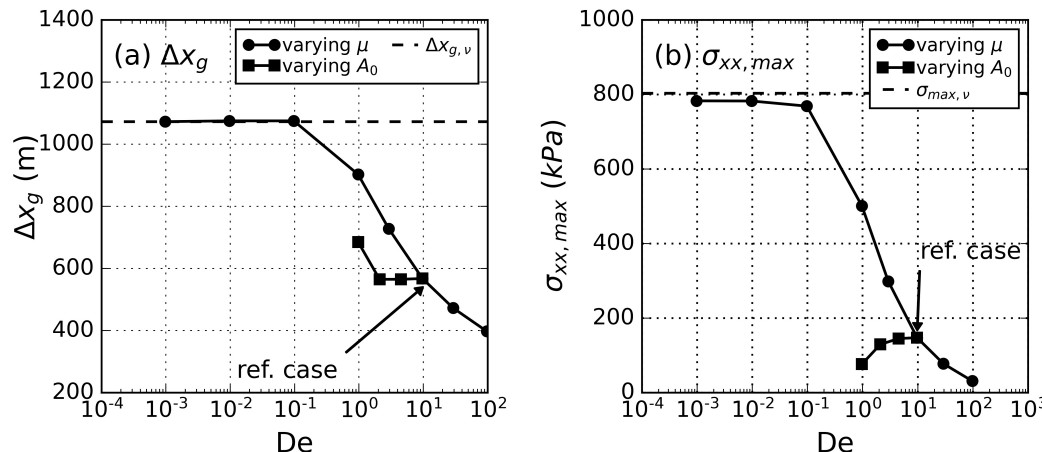

**Figure 4. (a)** The grounding-zone width $\Delta x_g$ (solid lines), defined as $\Delta x_g = \max\{x_r\} - \min\{x_l\}$ as a function of $De$. We vary $De$ by using two different schemes: (1) varying $\mu$ (round dots) from $\mu = 3 \times 10^7$ (right) to $3 \times 10^{12}$ Pa (left); (2) varying $A_0$ (square dots) from $1.2 \times 10^{-25}$ (right) to $1.2 \times 10^{-22}$ Pa$^{-3}$s$^{-1}$ (left). The dashed line is $\Delta x_{g,\nu}$, the grounding-zone width in the viscous limit ($\mu \to \infty$, $A_0 = 1.2 \times 10^{-25}$ Pa$^{-3}$s$^{-1}$). **(b)** Maximum tensile stress $\sigma_{xx,max}$ versus $De$ through a varying $\mu$ (round dots) or a varying $A_0$ (square dots). The dashed line is the tidal stress calculated by the viscous model (Stubblefield et al., 2021). The numerical reference case (Fig. 2) is labelled in both panels.

## 3.2 Sensitivity analysis

The viscoelastic model can be applied to other glaciers with different rheological properties and bed topography. Here we analyse the model's sensitivity to ice viscoelasticity and bed topography. From this point forward, we use $\sigma_{xx,max}$ to denote the low-tide maximum tensile stress within the lake region, which is assumed to directly contribute to hydrofracturing.

### 235   3.2.1  Sensitivity to the Deborah number

For a viscoelastic grounding zone, the tidal response is governed by the Deborah number ($De$), a dimensionless parameter defined as $De = \lambda f$, where $\lambda$ is the Maxwell time of ice and $f$ is the tidal frequency. Depending on $De$, the tidal response is primarily elastic ($De \gg 1$) or viscous ($De \ll 1$). Here, we explore the tidal response across a range of $De$ values from $10^{-4}$ to $10^2$, capturing the transition from viscous to elastic tidal response.

We vary $De$ across two different parametrization schemes: modifying the shear modulus $\mu$ or the prefactor $A_0$ in Glen's flow law. The variation in $\mu$ accounts for crevasses and damage that weaken the ice shelf, while changes in $A_0$ represents thermally controlled viscosity variations. All other parameters remain fixed at their reference values. We define a grounding-zone width, $\Delta x_g$, as the range of GL migration in a tidal period. In Fig. 4a, $\Delta x_g$ is plotted versus $De$. With the bedslope $\theta = 2 \times 10^{-2}$, a single GL is observed to migrate 0.5–1 km per tidal cycle. As $\mu \to \infty$ (with $A_0 = 1.2 \times 10^{-25}$ Pa$^{-n}$s$^{-1}$), the tidal response 245   becomes purely viscous, and the grounding-zone width converges to its viscous limit $\Delta x_{g,\nu}$, exceeding the elastic regime. By

contrast, increasing $A_0$ has a minimal effect on $\Delta x_g$. In Fig. 4b, as $\mu \to \infty$, the low-tide maximum tensile stress within the lake region $\sigma_{xx,max}$ increases and converges to the viscous stress calculated by the viscous model (Stubblefield et al., 2021). By contrast, as $A_0$ increases, the tidal stress slightly decreases due to the reduction in viscosity.

The Deborah number of ice is crucial in determining GL migration and tidal stress because a dominantly viscous response tends to increase the width of a grounding zone and alter the magnitude of tidal stresses, compared with the elastic limit. Since the tidal period is well constrained, our model indicates the importance of using viscoelasticity with an appropriate Maxwell time to predict the magnitude of tidal stress. Given the dependence of $\Delta x_g$ on $De$ (Fig. 4a), it may be possible to infer ice mechanical properties from observations on the range of GL migration.

### 3.2.2 Sensitivity to bedslope angle

The above discussion shows how tidal response varies with shear modulus, with a characteristic bedslope $\theta = 2 \times 10^{-2}$ in the Amery Ice Shelf grounding zone. Here we extend the results to different bedslopes and explore how the tidal response of a GL would change with local bathymetry. We consider three marine ice sheets with bedslope $\theta = 2 \times 10^{-4}$, $2 \times 10^{-3}$, and $2 \times 10^{-2}$, with all other parameters set to be the same as the reference case.

To simplify, we focus on the effect of $\theta$ while keeping the inflow velocity $u_0$ and the basal friction coefficient $C$ in Eq. 15 fixed. Because of this, the ice adjusts to the changing bedslope through either thinning or thickening. The modelled surface velocity near the GL deviates from the observed value $u_0$, but maintains the same order of magnitude. The GL migration is shown in Fig. 5a. Different from the single GL shown above, the low-bedslope regime $\theta = 2 \times 10^{-4}$ is characterized by double GLs at low tides. Between the left GL at $x_l$ and the right GL at $x_r$, the ice sheet is lifted due to a water layer trapped underneath, forming a "low-tide grounding zone" (Stubblefield et al., 2021). For the other two cases, only a single GL is found, with the range of the GL decreasing for increasing $\theta$. Moreover, the maximum tidal stress monotonically increases with increasing $\theta$ (Fig. 5b). For a specific GL, the local basal topography and characteristic bedslope angle can be constrained by observations (Fretwell et al., 2013). Thus, the uncertainties of the modelled tidal GL migration and stress mainly come from the rheological model.

### 3.3 Linear Elastic Fracture Mechanics model of the hydrofracture

The viscoelastic marine-ice-sheet model enables estimation of maximum tidal stress magnitudes within the grounding zone. In this section, through application of a fracture mechanics framework, we demonstrate how this calculated tidal stress might drive surface hydrofracture propagation. Since hydrofracturing typically occurs on a timescale short enough that the surrounding ice behaves elastically, we consider fracture propagation in the LEFM framework. The hydrofracture is assumed to be a quasi-static elastic fracture occurring at the location with $\sigma_{xx,max}$ at low tides. The stress that drives its propagation is the sum of the water pressure and tidal stress. The water pressure in the fracture $p_w$ is assumed to be hydrostatic; the tidal stress $\sigma_{xx,max}$ is calculated by the viscoelastic model mentioned above. We use the weight-function method to calculate the stress intensity factor $K_I$ (Tada et al., 2000). Since at low tides the GL goes downstream and leaves the ice beneath the lake attached to the

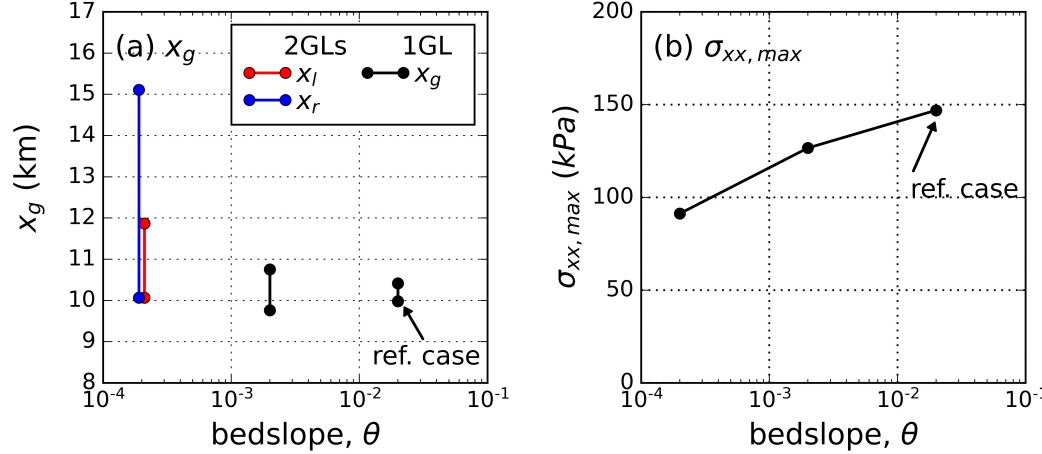

**Figure 5. (a)** The range of the GL position in one tidal period as a function of bedslope angle $\theta = 2 \times 10^{-4}, 2 \times 10^{-3}, 2 \times 10^{-2}$. When $\theta = 2 \times 10^{-4}$, there are two GLs. The left and right GLs are denoted $x_l$ and $x_r$, respectively. The other two cases give single GL $x_g$ shown by the black line. **(b)** Maximum tidal stress $\sigma_{xx,max}$ versus $\theta$. The numerical reference case in Fig. 2 is labelled.

bedrock, we use a weight function that is designed for ice grounded on rigid bedrock, as suggested by Jimenez and Duddu (2018). Details about the calculation of $K_I$ are provided in Appendix C.

Fig. 6a shows a schematic of the fracture model. The lake basin has a depth $d_b$ and is filled with water to a depth $d_w$. The horizontal stress $\sigma_{xx}(z)$ represents the low-tide tidal stress, and is obtained from the numerical results with a given tidal amplitude $A$. For a vertical fracture, we can calculate the stress intensity factor $K_I$ as a function of its length $d_l$. If $K_I$ exceeds the ice fracture toughness $K_C$, the fracture can propagate until $K_I = K_C$. We assume that lake drainage occurs when a vertical hydrofracture reaches the ice-sheet bottom.

Note that for the initial fracture, $K_I$ is sensitive to its length $d_{l,init}$. However, there is little observational constraint on the lengths of pre-existing fractures at lake basins. While the choice of $d_{l,init}$ requires further study, the relative importance of the lake pressure versus the tidal amplitude is independent of $d_{l,init}$, which is shown in the model-based criterion below. Here we will choose $d_{l,init}$ such that the model criterion best fits the drainage data from Trusel et al. (2022). For an initial fracture with $d_{l,init} = 0.1$ m to propagate, the critical tensile stress is approximately $\sigma_{xx} = 150$ kPa. For $d_{l,init} = 0.2$ m, the critical stress

is about 100 kPa, which cannot be achieved by background extension alone in the reference case shown in Fig. 2.

Fig. 6b shows a reference case with $K_I/K_C$ versus the fracture length $d_l$, utilising the vertical stress distribution ($\sigma_{xx}$) derived from numerical simulations in subsection 3.1. We evaluate different combinations of lake depth $d_w$ ($d_w = 0, 2, 4$ m) and tidal phases (low tide and high tide). The vertical dashed line represents the ice fracture toughness $K_C$. During low tide, downward flexure generates positive $K_I$ near the ice surface, whereas high-tide compressive stresses produce negative $K_I$ at

the same location. Fig. 6b can be used to predict lake drainage: at low tides, for a pre-existing fracture with length $d_{l,init}$, if $K_I > K_C$ holds for any depth that the fracture can reach, then $A$ and $d_w$ are predicted to induce lake drainage. By iteratively

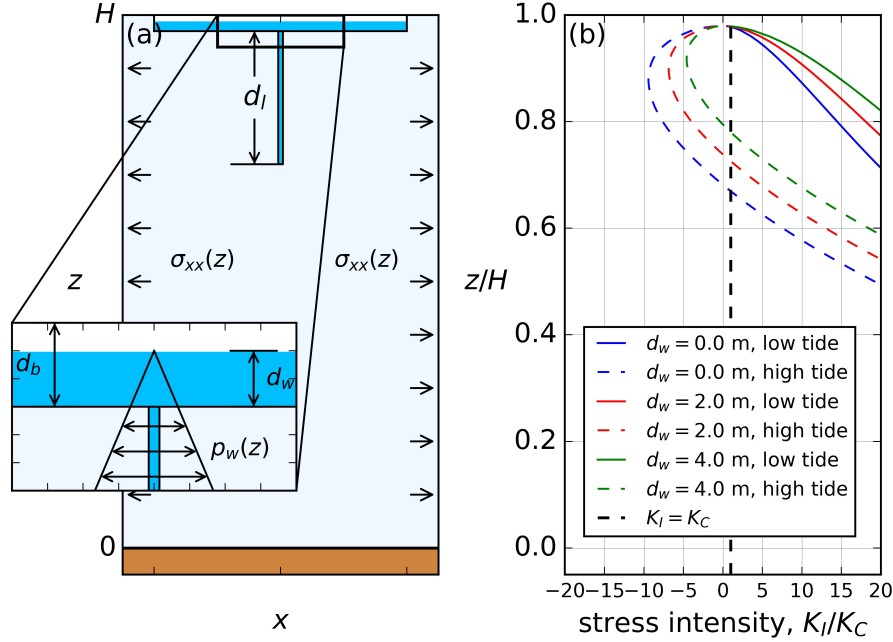

**Figure 6. (a)** The LEFM model of hydrofracture. The lake basin with depth $d_b$ is filled with water to a depth $d_w$. Here $d_w$ serves as a measurement of the water pressure $p_w$, as shown in the zoom-in window. Promoted by the tidal stress $\sigma_{xx}(z)$ and lake-water pressure $p_w$, a vertical fracture with length $d_l$ is initiated from the lake bottom. **(b)** A reference case showing $K_I/K_C$ varying with depth (scaled by the ice thickness) and tidal phases, with $A = 1.0$ m, $d_b = 10$ m, and $d_{l,init} = 0.1$ m. The solid lines represent $K_I/K_C$ at low tides. The dashed lines represent $K_I/K_C$ at high tides when upward flexure causes compression, and thus a negative tidal contribution to hydrofracture.

applying this criterion across combinations of $A$ and $d_w$, we assess hydrofracture propagation likelihood for ranges of $A$ and $d_w$ values. This establishes a model-based drainage criterion within a two-dimensional parameter space defined by tidal amplitude ($A$) and lake depth ($d_w$). We discuss this model-based drainage criterion and its application to lake drainage in the Amery grounding zone in subsection 4.2.

## 4 Discussion

In Sect. 3, we use a viscoelastic model to predict GL migration and tidal stress of a marine ice sheet, which is then integrated in a LEFM framework to assess tidally-induced hydrofracturing. For the grounding zone analysed in Trusel et al. (2022), tidal responses could be more complex than the idealised flow-line model due to lateral stresses. Nevertheless, to our knowledge, the Trusel et al. (2022) dataset is the only observational record of tidally induced lake drainage that enables a quantitative analysis. In the following section, we first derive the strain-rate field near the lake studied by Trusel et al. (2022) using remotely sensed ice-surface velocity data. Then, we construct a model-based drainage criterion for laterally unconfined grounding zones, explicitly neglecting contributions from lateral stresses, and test its predictions against the observations of Trusel et al. (2022).

While the model-based criterion yields a prediction closely aligned with data, deviations are observed. Finally, we discuss the limitations of our model that could cause the deviations, and propose potential refinements to extend the model's capability to laterally confined glaciers, which are important in governing the upstream ice-sheet mass balance.

## 4.1 Application to the Amery grounding zone: ice flow field and lake-basin bathymetry

Trusel et al. (2022) report drainage characteristics for a supraglacial lake located at 70.59 °S, 72.53 °E. Fig. 7(a) shows the ice-surface velocity field $v$ from the MEaSUREs InSAR-Based Antarctica Ice Velocity Map (Rignot et al., 2011b, 2017; Mouginot et al., 2012, 2017) in this region, from which we calculate strain rates $\dot{\varepsilon}$ and the ice-flow line that bisects the lake location. The lake is located on the GL position given by the MEaSUREs Antarctic Grounding Line from Differential Satellite Radar Interferometry, Version 2 (Rignot et al., 2016b, 2011a, 2014; Li et al., 2015). We use BedMachine Antarctica v2 to obtain the basal topography and ice geometry along the flow line (Fig. 7b) (Morlighem et al., 2017, 2020). The subglacial cavity downstream of the grounding zone is more than 20 m in height, which is large enough to allow free tidal oscillation without the formation of pinning points. Thus, we assume that the ice shelf downstream of the GL does not contact the bedrock, and that the water pressure on the ice–ocean interface is hydrostatic. In the computation, we use a linear bedrock topography (Fig. 7b) with a slope angle chosen to approximately match observations. In Appendix B we present model results with real bed topography for comparison.

Following Wearing (2017), we decompose the ice strain rate into an along-flow component and a transverse component. For each grid point, we compute the ice velocity orientation, defining the along-flow and transverse directions using unit vectors $\hat{v}$ and $\hat{t}$, respectively, within a local right-handed coordinate system. The along-flow strain rate is then derived pointwise as

$$\dot{\varepsilon}_p = \hat{v} \cdot \dot{\boldsymbol{\varepsilon}} \cdot \hat{v}, \tag{18}$$

and the transverse strain rate is

$$\dot{\varepsilon}_t = \hat{v} \cdot \dot{\boldsymbol{\varepsilon}} \cdot \hat{t}. \tag{19}$$

Fig. 7(c,d) shows $\dot{\varepsilon}_p$ and $\dot{\varepsilon}_t$ near the lake. The background stress is dominantly extensional away from the GL. However, near the Amery Ice Shelf shear margin, ice flow deflects rightward, inducing a transverse strain-rate component. Meanwhile, upstream of the GL, the outlet glacier undergoes lateral shear that could modulate the along-flow momentum balance and the GL position. In our model, we focus on the extensional stress along the flow line and neglect factors related to lateral stresses.

To estimate the contribution from lake water supply to hydrofracturing, we need to estimate the hydrostatic pressure at the bottom of the lake basin. To this end, we obtain the elevation of the basin from a 2-m resolution WorldView-1 DEM captured when the lake was dry. The elevation of the flat basin, excluding any craters or hydrofractures, is considered the lowest point of the lake. The basin is approximately 10 m deeper than the surrounding terrain and spans about 1 km in the direction perpendicular to the GL. The lake-water depth prior to each drainage event is calculated by subtracting the lake-basin elevation from the median shoreline elevation, which we obtain using a shoreline-extraction technique as outlined in Moussavi et al. (2016) and Trusel et al. (2022).

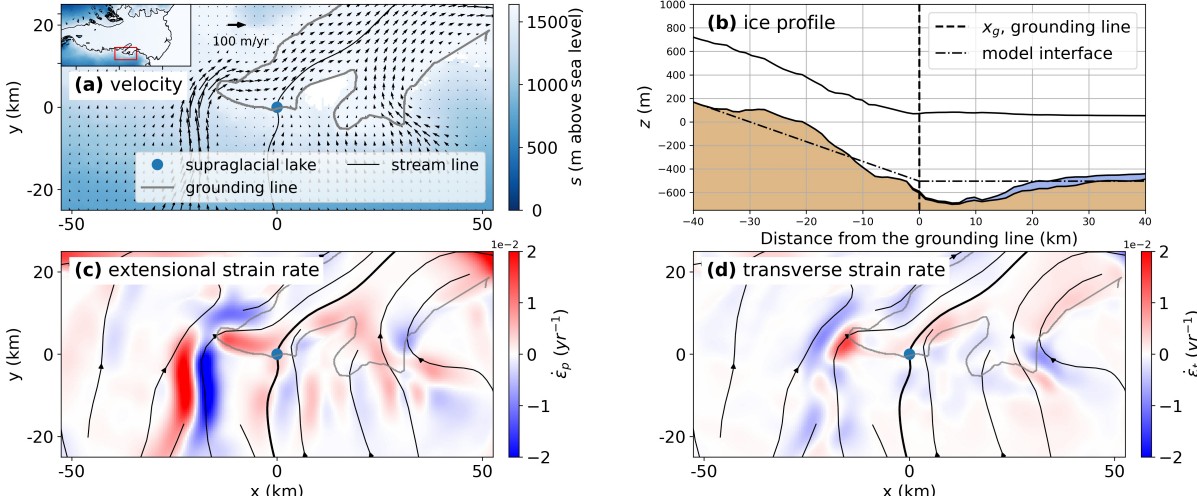

**Figure 7.** Ice surface velocity and strain rate in the region of the supraglacial lake. **(a)** Velocity field near the grounding line (grey), where the supraglacial lake is denoted with the blue dot. The solid black line is the streamline flowing through the lake location. The colour represents the ice-sheet surface elevation above sea level. The map inset at the top left corner shows ice-surface elevation for the full Amery Ice Shelf, with the plotted region outlined with a red box; **(b)** ice-sheet geometry and bed topography along the streamline in panel **a**. Note that $x = x_g = 0$ is the position of the supraglacial lake as well as the grounding line. The dash-dotted line is the idealised ice-bottom profile used in the model; **(c)** along-flow strain rate with positive values representing extension; **(d)** transverse strain rate.

## 4.2 Application to the Amery grounding zone: drainage criteria from tidal amplitude and lake depth

In subsection 3.3 we introduced how we construct the model-based criterion in the two-dimensional parameter space defined by tidal amplitude ($A$) and lake depth ($d_w$). The criterion corresponds to the marginal threshold at which an initial fracture propagates to the ice–bedrock interface. In Fig. 8a, we show two such criteria (denoted by square markers) for initial crack lengths of $d_{l,init} = 0.1$ and $0.2$ m. When the parameter pair $(A, d_w)$ of a lake crosses the criterion from bottom-left to top-right, the total tensile stress becomes large enough to enable hydrofracture penetration to the ice-sheet base. An increasing $A$ reduces $d_w$ along the criteria. This inverse relationship quantifies the relative importance of tidal flexure to water pressure in driving lake drainages through hydrofracturing. It suggests that, in grounding zones of laterally unconfined ice shelves, ocean tides cause tidal flexure that reduces the lake depth required for drainage through hydrofracturing.

Furthermore, the criterion identifies a critical $A$ beyond which fracture propagation can occur solely due to tidal flexure, and hence independently of water supply. When $A$ is beyond this threshold, supraglacial lakes are unlikely to form because tidally driven flexure would open fractures in the grounding zone. Such fractures may act as vertical conduits, transporting surface water to the bed and preventing the meltwater accumulation that forms supraglacial lakes. These hypotheses could be tested by a study of more supraglacial-lake drainage events with tidal amplitude measured locally.

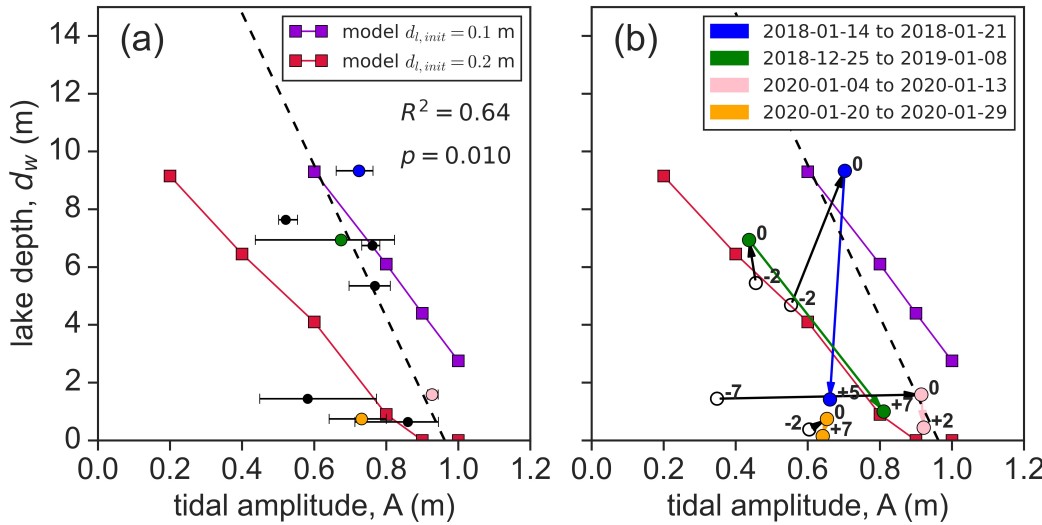

**Figure 8. (a)** A comparison between the model criterion and the drainage data. Each circle represents one drainage event from Trusel et al. (2022). The horizontal coordinate is the time-averaged daily maximum tidal amplitude during the drainage, with an error bar representing the range of the daily maximum tidal amplitude. The vertical coordinate is the pre-drainage lake depth. The dashed black line is a weighted linear regression of the observations. The red and violet lines are model-based criteria with different initial crack lengths, with squares representing the numerical experiments. The four coloured circles represent drainage events with best-constrained temporal evolution of lake depth and tidal amplitude. **(b)** Temporal evolution of coloured events in panel **a**. The points labelled "0" represent the day of the drainage. The negative and positive values represent the days before and after the drainage, respectively.

In Fig. 8a, we compare the model-based criterion with data from Trusel et al. (2022). The data cluster close to the model-based criterion for $d_{l,init} = 0.1, 0.2$ m. A weighted linear regression of the observations suggests that a higher tidal amplitude reduces the lake depth required for drainage, indicating the importance of tidal flexure in driving hydrofracturing, as shown in our model. However, the steeper slope of the regression line suggests the dependence of lake drainage on tidal amplitude is stronger than predicted by our model.

To better demonstrate the drainage process, Fig. 8b shows four events that have the best observational constraint on temporal evolution of lake depth and tidal amplitude. We show measurements from before, during, and after the drainage. Note that we simply assume the drainage occurs at the highest water level observed, which is the minimum pre-drainage water level due to the time interval between satellite images. The two events dated 2018-12-25 and 2020-01-04 cross the criterion with $d_{l,init} = 0.2$ m during the observational interval of several days. The event dated 2018-01-04 crosses both criteria. The post-
drainage states are below the criterion with $d_{l,init} = 0.1$ m, representing the end of drainage due to insufficient water supply.

### 4.3 Limitations

Deviations between the model-based criterion and data regression indicate that we may be underestimating the tidal contribution to hydrofracturing (Fig. 8). As indicated above, this could be due to the model not including the effects of lateral stresses in the grounding zone, which could alter the tidal response. To address this in future work, the lateral shear stress can be parametrised by an additional term in the along-flow momentum equilibrium equation (Eq. 2),

$$\tau_w \sim \frac{|u|^{1/n-1}}{W^{1/n+1}}, \tag{20}$$

where $u$ is the along-flow ice velocity, and $W$ is the half-width of the glacier (e.g., Schoof, 2007a).

In addition, we assume a stress-free top surface in the viscoelastic flow model. However, the supraglacial lake can induce additional stress in the surrounding ice, particularly on floating portions of the grounding zone that allow downward flexure (MacAyeal and Sergienko, 2013). Meanwhile, the existence of ice fatigue due to stress oscillations can weaken ice strength and promote hydrofracturing (Borstad et al., 2012; Lhermitte et al., 2020). A better approach may be to consider the supraglacial lake and the ice damage directly within the 2-D viscoelastic model.

Another limitation arises from our assumption of hydrostatic water pressure on the ice–ocean interface. The pressure gradient induced by tidally modulated subglacial water flow can cause elastic flexure in ice sheets close to the grounding line (Warburton et al., 2020). Furthermore, ocean tides can change the effective pressure at the bed and lubricate the ice–bedrock interface, leading to a variation of basal friction that is not accounted for by the sliding law in our model (Gudmundsson, 2011). Thus, it is important to incorporate the subglacial hydrology in simulating the tidal response of a marine ice sheet.

Limitations also come from the data availability. Relative to the tidal period and lake-drainage period, the lower temporal resolution of the remotely sensed observations might obscure the true lake depth and tidal amplitude at the time of drainage. Field measurements and satellite images of supraglacial lake drainage with a higher observational frequency could improve our understanding of tidally-induced drainage.

### 5 Conclusions

Our study of tidally-induced stress and hydrofracture propagation in a laterally unconfined, viscoelastic, marine-ice-sheet grounding zone suggests that ocean tides can generate significant stress near the grounding line. These stresses can increase the vulnerability of ice sheets to hydrofracturing in grounding zones where lakes form. We further establish a model-based criterion for lake drainage that links ocean tides and lake depth to supraglacial lake drainage via hydrofracture. Importantly, the criterion indicates that tidal flexure reduces the critical lake depth required for drainage through hydrofracturing. While our model formulation simplifies the ice momentum balance by neglecting lateral shear stress, this study provides the first integrated assessment of tidally-induced viscoelastic grounding line dynamics and fracture propagation, which helps to explain the role of ocean tides in driving grounding line migration and supraglacial lake drainage in marine ice sheets.

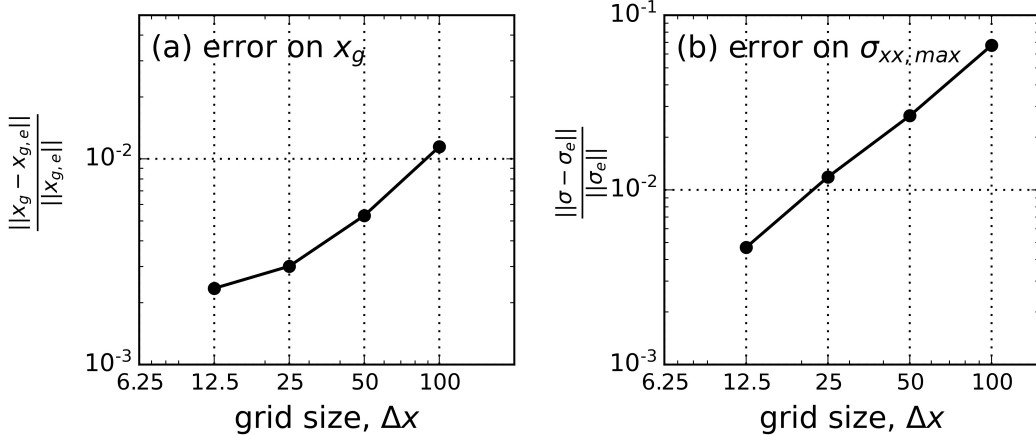

**Figure A1.** Convergence of (a) GL position and (b) maximum tensile stress $\sigma_{xx,max}$ with decreasing element size $\Delta x$ (12.5 m, 25 m, 50 m, 100 m). For simplicity, we denote $\sigma_{xx,max}$ by $\sigma$ without causing any confusion. Here $x_{g,e}$ and $\sigma_e$ denote the exact solution to the GL position and maximum tensile stress $\sigma_{xx,max}$, respectively. $||\cdot||$ is the $L^2$ norm.

*Code and data availability.* The code and data used for the reference case in Sect. 3 is available at the online GitHub repository https://github.com/HwenZhang/TidalHydroFrac.git. It can be modified to reproduce the results presented in Sect. 4 using the values of parameters provided in the text.

## Appendix A: Convergence test

The convergence test shows the results are mesh-independent. Here, we apply the same flow law parameter values with $A_0 = 3.2 \times 10^{-24} \mathrm{Pa}^{-3}\,\mathrm{s}^{-1}$ and $n = 3$ as (Stubblefield et al., 2021). Considering a marine ice sheet with bedslope $\theta = 10^{-3}$ and friction coefficient $C = 7 \times 10^5$, we use the fine-grid solution $x_{g,e}$, $\sigma_e$ ($\Delta x = 6.25\mathrm{m}$) as the exact solution. All other parameters are kept same as the reference case. Here $x_{g,e}$ denotes the time series of the exact GL position, and $\sigma_e$ denotes the time series of the exact maximum tensile stress on the ice-sheet surface within the lake region $|x - \bar{x}_g| \leq 0.5$ km. As $\Delta x$ decreases, the GL position $x_g$ and maximum tensile stress $\sigma_{xx,max}$ linearly converge to the fine-grid solution (Fig. A1).

## Appendix B: Simulation with real bed topography

In Fig. B1 we present model results using the real bed topography shown in Fig. 7b (Morlighem et al., 2017, 2020). For comparison, the physical properties of ice and basal slipperiness $C$ are kept the same as in idealised models with a linear bed. The jagged variation of $x_g(t)$ is a consequence of the use of coarse grids near the grounding line for convergence. While the tidal stress and grounding-zone width are modified by bed undulation, the results have an equivalent order of magnitude to the

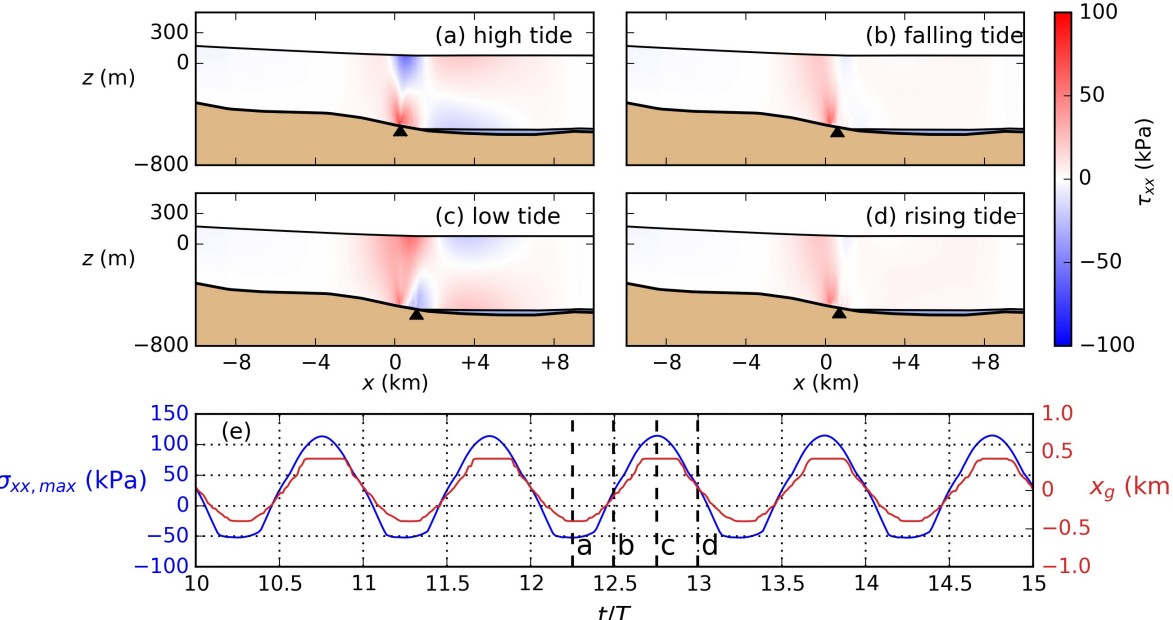

**Figure B1.** Tidal response of the the ice shelf with real bed topography around the lake reported in (Trusel et al., 2022). **(a)–(d)** Deviatoric tensile stress $\tau_{xx}$ in one tidal period. **(e)** The maximum tensile stress $\sigma_{xx,max}$ (blue) on the top boundary within the lake region ($\bar{x}_g - 0.5$ km $\leq x \leq \bar{x}_g + 0.5$ km) and the GL position $x_g$ (red) versus time (scaled by the tidal period $T$) with positive values representing downstream migration. Vertical dashed lines show the time of panels (a)-(d).

case with linear bed topography, indicating the importance of tidal stress regardless of bed roughness. Therefore, we use the linear bed topography in the model.

## Appendix C: Weight function method

In this appendix, we introduce the weight function method to calculate the stress intensity factor $K_I$ of a vertical surface hydrofracture of length $d_l$. Fig. C1 shows the schematic of the fracture and weight function method. The stress contributing to hydrofracturing is denoted as $\sigma_f = \sigma_{xx} + p_w$, which combines the net stress $\sigma_{xx}$ from the model with the internal water pressure $p_w(z)$ within the fracture. Assuming $p_w(z)$ is hydrostatic, it is given by $p_w(z) = \rho_w g (d_l + d_w)$, where $d_w$ is the water depth of the lake. Here the ice stress $\sigma_{xx}(z)$ represents the vertical stress distribution at the location where the maximum surface horizontal tensile stress $\sigma_{xx,max}$ is observed.

The factor $K_I(d_l)$ is then computed by integrating $\sigma_f$ along the fracture using a weight function $G_1$ (Tada et al., 2000),

$$K_I(d_l) = \int_{h_0 - d_l}^{h_0} \sigma_f(z) G_1\left(\frac{d_l}{H}, \frac{h_0 - z}{d_l}\right) dz, \qquad (C1)$$

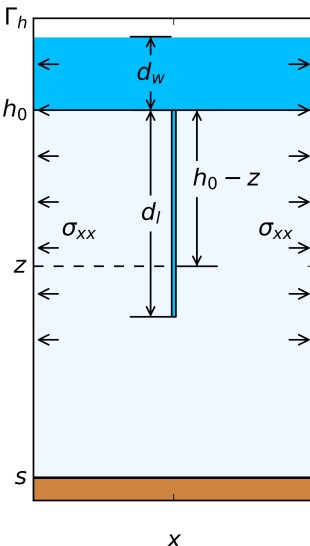

**Figure C1.** Schematic of the fracture geometry used in the weight function method.

where $h_0 = \Gamma_h - d_b$ is the lake basin elevation, $d_l/H$ is the fracture length scaled by local ice thickness $H$ and $(h_0 - z)/d_l$ denotes the depth below the lake basin, scaled by $d_l$. The weight function $G_1$ accounts for the surface hydrofracturing within an ice sheet located on a rigid bedrock (Jimenez and Duddu, 2018),

$$
\qquad G_1\left(\lambda, \gamma\right) = \frac{2}{\sqrt{2H}} \sqrt{\frac{\tan\left(\frac{\pi\lambda}{2}\right)}{1 - (\cos\frac{\pi\lambda}{2}/\cos\frac{\pi\lambda\gamma}{2})^2}} \left\{ 1 + 0.3\sqrt{1 - \gamma^{5/4}} \left[ 0.5\left(1 - \sin\frac{\pi\lambda}{2}\right)\left(2 + \sin\frac{\pi\lambda}{2}\right)\right] \right\}. \tag{C2}
$$

*Author contributions.* **Hanwen Zhang**: Conceptualization, Methodology, Software, Validation, Formal analysis, Investigation, Data Curation, Writing - Original Draft, Writing - Review & Editing, Visualization. **Richard Katz**: Conceptualization, Software, Resources, Writing - Review & Editing, Supervision, Project administration, Funding acquisition. **Laura Stevens**: Conceptualization, Resources, Writing - Review & Editing, Supervision, Project administration.

*Competing interests.* The authors declare that they have no conflict of interest.

*Acknowledgements.* The authors thank L. D. Trusel and A. Fatula for help in interpretation of their lake-drainage observations, and I. Hewitt, C.-Y. Lai, and the RIFT-O-MAT group for discussions on grounding line dynamics and model set-up. We acknowledge three anonymous reviewers for their constructive comments that substantially improved this manuscript. This research received funding from the European Research Council under Horizon 2020 research and innovation program grant agreement number 772255.

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
