# Peer review of "Viscoelastic mechanics of tidally induced lake drainage in the grounding zone"

_EGUsphere, 2024_

## Referee Comment (RC1)

**Review of egusphere-2024-665**

*General comments:*
The manuscript "Viscoelastic mechanics of tidally induced lake drainage in the Amery grounding zone" by Zhang et al. explains the physics that drive a series of supraglacial lake drainage events in Antarctica with numerical models. Remote sensing data suggest that the extensional stress regime of the background ice flow is not enough to trigger the lake drainage events. They conduct a series of targeted numerical experiments to show that tidal flexure provides the necessary extensional stresses to drive hydrofracturing, depending on the depth of the supraglacial lake. This essentially confirms the hypothesis in the observational study, Trusel et al. (2022), that detailed these drainage events. While I have some comments primarily related to clarification and discussion, my judgment is that this would be an excellent contribution to TC.

*Specific comments:*
- In the last line of the abstract and conclusions you mention calving. While I understand that supraglacial lakes might play a role in ice-shelf breakup (Banwell et al., 2019), I was not exactly sure how the results in this study related to the calving front. Clearly there are similar physics because you are modelling fracture and flexure, but additional clarification would be helpful if you want to include this statement.
- Introduction: Maxwell time of "approximately 9 hours in our estimation", this needs some more context for how you calculated this or a reference.
- Section 2.1, last paragraph, "In B" should be "In Appendix B"?
- When you introduce the upper-convected derivative, you should add a reference and probably provide some motivation (i.e. objectivity). The following review is excellent:
    Snoeijer J. H., Pandey A., Herrada M. A. and Eggers J. (2020). The relationship between viscoelasticity and elasticity. Proc. R. Soc. A. 47620200419
- Regularized flow law: It is good that you explicitly discuss the regularization because this is often not the case. I think it might be worth noting, with proper references, that the ice viscosity can vary over several orders of magnitude, and that the upper bound you have set seems to be on the lower end of the spectrum? Also suggest adding a statement here that you later test sensitivity to the Maxwell time.
- Comment somewhere on how the modelled grounding zone widths compare to width estimated from interferometry (Chen et al., 2023)?
- After equation (18), "…variational formulation weakly converges…". To my knowledge, this is less obvious for the UCM model because it cannot be cast directly as a minimization problem like those dealt with in Kikuchi & Oden (1988). Nevertheless, approximating the contact conditions by $\sigma_e = \max\left(0, \sigma_e + \frac{1}{\epsilon} u_n\right) \approx \max\left(0, \frac{1}{\epsilon} u_n\right)$ for small $\epsilon$ (where $\sigma_e = \sigma_n - p_w$) still makes sense for UCM and motivates the use of a penalty term, as long as there aren't singularities in $\sigma_e$.
- Last paragraph of section 2.5, "In A" should be "In Appendix A"?
- Section 2.6 seems kind of random at first glance and needs more context… e.g., say what are you going to do later with the lake depths? Also, it might be better to place this after Section 2.1 rather than after the modelling material.

- Table 1: Units on viscosity and friction regularization parameters?
- Section 3.1: Specify that $\sigma$– A relationship is for $\sigma_{xx,max}$.
- Figures 5 and 6 needs to label panels (a) and (b)
- Figure 5: clarify why the dashed lines go into the positive region? I thought they were compressive at high tide so not contributing to fracture, and I became *confused*.
- Section 3.2: Not many details are provided about the weight function method. I presume that you are doing something with $\sigma_{xx,max}$ but some more context would be helpful.
- I don't think you say what is the value of fracture toughness $K_C$ ?
- Section 3.3: All of this is in terms of the stress intensity factor, but I was wondering what are the stress thresholds associated with fracture propagation so that you can relate these to the background extensional stress (<40 kPa), which you say is not enough to cause fracture propagation on its own? Something to verify this claim would be good.
- Section 3.3: "Supraglacial lakes would not be able to form under such large tidal stress." Does this limit correspond to the zero water depth in Figure 6 (state if so)? This is also an interesting point that you could revisit in the discussion.
- Section 4: You are using present tense "We use" / "We construct" but you have already done these things at this point in the paper so maybe "We used" or "We have used"?
- Section 4.1: Change "Ice Maxwell time" to "The Maxwell time of ice"?
- Section 4.1: I thought $\lambda$ was the Maxwell time but here you are using $\tau$, which was already used for deviatoric stress?
- Section 4.1: "may be possible to infer ice mechanical properties from observations on the range of GL migration". This is a really interesting idea that could be discussed a little more, e.g., reference back to grounding zone width ranges from interferometry?
- Section 4.3: "is important to incorporate the subglacial hydrology…". I felt like this should have some references. Also in previous paragraph, should include a ref. for "damage".
- Section 4.3: "the supraglacial lake can induce additional stress in the surrounding ice". In Section 3, you neglect the influence of the lake, but here you say that it may be important. Maybe say in Section 3 that "we later revisit this assumption…" or similar?
- Sensitivity tests: Does it matter that you are varying the elastic modulus to change the Maxwell time instead of the viscosity? In other words, would you get a different result if you instead varied the viscosity at fixed shear modulus?
- Discussion: General comment. I felt like some of the implications of the results could be discussed more, as noted in a couple instances above. As it stands the discussion is sort of technical with the focus on sensitivity tests. The sensitivity tests provide valuable information but I think they could be placed in a broader context – e.g., how do shear modulus, bedslope at the GL, or grounding zone width vary in Antarctica, and what implications does this have in light of the tests? For example, it seems from Figure 8b that lower bedslopes have lower stresses and therefore less likely for lake drainage, but this isn't discussed? Does having a wider grounding zone at lower bedslope have any influence on the lakes? Also, are there any broader implications with respect to rifting/stability/vulnerability of ice shelves, given that you talk about these things in the introduction? These are just examples that could lead to a more stimulating discussion.

---

## Referee Comment (RC2)

**Review of "Viscoelastic mechanics of tidally induced lake drainage in the Amery grounding zone" by Zhang et al.**

The authors use a two-dimensional model (the vertical cross-section) to simulate ice flexure at the grounding zone caused by diurnal tides with the goal to explain drainage of a supraglacial lake on the Amery Ice Shelf observed by Trusel et al. (2022). The authors begin with estimates of stress regime of the outlet glacier and the grounding zone where the draining lake was observed and then proceed to describe their model and its results.

I have two fundamental problems with this study. The first one is application of the results of a highly idealized study to the observed lake drainage. Although I commend the authors for trying to estimate various stresses from remote sensing observations, the estimates seem to have been done after the fact, as an attempt to justify the modeling assumptions. These estimates, or rather presented or describe data suggest that various assumptions made in this study are violated. This suggests to me that conclusions drawn from the model results are not applicable to the stress regimes and conditions at the outlet glacier and the grounding zone of the Amery Ice Shelf. Firstly, the authors argue that because the background stresses are small, they "focus on the the streamline and adopt a 2-D flow line model" (as a side comment, it would be highly desirable for the manuscript to have lines numbered). The figure below is fig. 1a of the manuscript. It shows

[Figure]

Figure 1: Panel (a) of fig.1 of the manuscript. The thin red line depicts how ice velocity $u$ varies along the width of the outlet glacier $W$,

that the flow of the outlet glacier is strongly affected by the lateral shear.

The size of the arrows depicting the ice velocity are much lower outside of the centreline. My hand-drawn thin red line meant to illustrate how the along the flow ice-velocity component changes across the outlet glacier. This is a typical velocity profile of ice flow strongly affected by the lateral shear caused by the presence of the lateral confinement (*e.g. Raymond, 1996*). Its effects cannot be ignored. Consequently, they need to be accounted for either by having a three-dimensional model that includes the second horizontal dimension transverse to the ice flow and imposing the relevant conditions on the lateral boundaries, or by parameterizing them in the momentum balance eqn. (3). These effects of the lateral shear will substantially alter the model results.

Secondly, the quoted magnitude of the observed velocity imposed at the inflow boundary is low, 9 m/yr; so are the magnitudes of velocity shown in fig. 1f (a minor comment: it is unclear whether this velocity profile is computed or observed). Using parameters listed in table 1 and the Shallow Ice Approximation one could estimate the ice surface velocity resulted from the internal deformation only, assuming no-slip at the ice bed interface. That value is $\sim$20 m/yr, which is larger than the observed surface velocity by a factor of two. This suggests that (a) either the chosen parameters are off (specifically the ice stiffness parameter $A_0$, which I will come back to) or (b) the ice flow is dominated, or strongly influenced, by the vertical shear, and the focus on the longitudinal stress $\tau_{xx}$ is unwaranted, or both.

Thirdly, the chosen value of $A_0$ is very high. The ice-stiffness parameter is a function of the temperature of ice through its column. The chosen value would correspond to ice temperatures of the range from -5°C to -7°C, which is very warm. Although summer temperatures can exceed freezing point from time-to-time, as indicated by the supraglacial lakes, the annual mean surface temperature is around -20°C (*e.g.*, Kittel et al., 2021). With ice flow primarily driven by the internal deformation, the ice temperature through the most of the ice column is not substantially warmer; it is only in the fairly narrow band near the bed it is warmer due to the geothermal heat flux. The very high chosen value of the ice stiffness parameter leads to a very low ice viscosity, of the order of $10^{13}$ Pa·s, which is at least an order, or more likely two orders of magnitude lower than the typical values of ice viscosity.

This brings me to the second problem with the study — the choice of the ice rheology. The authors have estimate it 9 hrs (the penultimate line on page 2) and 40 hrs (the penultimate line of section 2.3 page 6). For more realistic values of ice viscosity it is of the order 5-15 days, which

is substantially longer than the period of diurnal tides that cause the ice flexure. This fairly unambiguously indicates that ice responds to diurnal tides as elastic medium. Two questions that immediately comes to mind — is it worth the effort the authors have gone through and complexity of the viscoelastic rheology? Can't one simulate it with much simpler elastic rheology?

Considering all issues with the study, applications of its results to the lake drainage on the Amery Ice Shelf and the drawn conclusions are questionable. It is entirely plausible that tides might have played a role in it. However, one cannot make any relevant statements based on the presented results.

**References**

Kittel, C. et al. Diverging future surface mass balance between the Antarctic ice shelves and grounded ice sheet. *Cryosphere* 15, 1215–1236 (2021).

Raymond, C (1996) Shear margins in glaciers and ice sheets. *Journal of Glaciology* 42(140), 90–102. doi:10.3189/S0022143000030550

Trusel, L. D., Pan, Z., & Moussavi, M. (2022). Repeated tidally induced hydrofracture of a supraglacial lake at the Amery Ice Shelf grounding zone. *Geophysical Research Letters*, 49, e2021GL095661. https://doi.org/10.1029/2021GL095661

---

## Referee Comment (RC3)

**Review of egusphere-2024-665**

This paper aims to estimate the importance of tidally-driven fracture propagation in a supraglacial lake in the Amery grounding zone. Using an idealized flowline simulation of Amery Ice Shelf, the deviatoric tensile stress at different times during the tidal cycle is estimated in a Full-Stokes model with a visco-elastic rheology. Linear Elastic Fracture Mechanics are used to estimate the stress intensity factor, and a threshold condition is used to estimate when lake drainage could occur. I found this an interesting study, proposing a plausible process for the observed lake drainage events. However, I have a few comments about the model setup and presentation that should be addressed before publication.

**1 Major comments**

1. I found the information provided in section 3.2 about the LEFM model inadequate – I don't think sufficient information is provided to reproduce the stress intensity factors shown in the study.

2. Part of the stress field seen in figure 3 is a result of keeping the ice thickness fixed: In full-Stokes models, when you have flow across a slip/free slip boundary, the ice surface adjusts to have a very characteristic dip just downstream of the grounding line, which is a result of the speed up across this boundary (e.g., Barcilon and MacAyeal, 1993; Nowicki and Wingham, 2008). If the surface cannot adjust, residual stresses at the surface occur. Ideally, simulations would have been done with an evolving surface, but I understand that this is beyond the scope of this study. However, to account for this limitation, the difference of the deviatoric surface stress with tides and without (i.e., for a static grounding line) should be used (which I guess would be about 10 kPa less than the stresses shown, judging from the figures 3b and d).

3. The existence of a 10 m deep lake at the ice surface imposes a pressure of about 10 kPa at the ice surface, which is not necessarily negligible. How would that alter the stress considerations?

**2 Minor comments**

1. Page 4, 2nd line: "the the" $\rightarrow$ "the"

2. Page 4, 2nd paragraph: "The subglacial cavities downstream of the grounding zone are more than 20 m wide." I was confused by this, as BedMachine does not have 20 m resolution. What are you referring to here?

3. Figure 1b and caption: there is reference here to $\sigma_1$ and $\sigma_2$ but elsewhere in the text you refer to principle strain rates. Please make sure this is consistent.

4. Figure 6 and corresponding text: can you comment on the zero-tidal amplitude limit? Are the results what you would expect?

**References**

V. Barcilon and D. R. MacAyeal. Steady flow of a viscous ice stream across a no-slip/free-slip transition at the bed. *Journal of Glaciology*, 39:167–185, 1993.

S. M. J. Nowicki and D. J. Wingham. Conditions for a steady ice sheet–ice shelf junction. *Earth and Planetary Science Letters*, 265(1):246–255, 2008.

---

## Author Comment (AC2)

**EGUSphere-2024-665 – Response to referee 1**

September 18, 2024

**1 General comments from the referee**

The manuscript "Viscoelastic mechanics of tidally induced lake drainage in the Amery grounding zone" by Zhang et al. explains the physics that drive a series of supraglacial lake drainage events in Antarctica with numerical models. Remote sensing data suggest that the extensional stress regime of the background ice flow is not enough to trigger the lake drainage events. They conduct a series of targeted numerical experiments to show that tidal flexure provides the necessary extensional stresses to drive hydrofracturing, depending on the depth of the supraglacial lake. This essentially confirms the hypothesis in the observational study [Trusel et al., 2022], that detailed these drainage events. While I have some comments primarily related to clarification and discussion, my judgment is that this would be an excellent contribution to TC.

We thank the reviewer for their constructive and insightful comments that have helped us improve the manuscript. Below we provide a detailed discussion of the comments and proposed changes. We use blue colour to indicate **comments**; our **replies** are in black.

**2 Specific comments from the referee**

In the last line of the abstract and conclusions you mention calving. While I understand that supraglacial lakes might play a role in ice-shelf breakup [Banwell et al., 2019], I was not exactly sure how the results in this study related to the calving front. Clearly there are similar physics because you are modelling fracture and flexure, but additional clarification would be helpful if you want to include this statement.

It was meant that the developed hydrofractures, if they remain open, can be advected downstream and destabilise the ice shelf by causing rifting. To avoid the confusion, we will remove the word "calving" and replace "rifting" with a general description "crevassing".

Introduction: Maxwell time of "approximately 9 hours in our estimation", this needs some more context for how you calculated this or a reference.

We will revise the paragraphs related to the Maxwell constitutive law. A detailed context for the revision can be found in the response to Referee 2. Below we provide details on how we choose the new set of rheological parameters and calculate the tidal stress.

Regularized flow law: It is good that you explicitly discuss the regularization because this is often not the case. I think it might be worth noting, with proper references, that the ice viscosity can vary over several orders of magnitude, and that the upper bound you have set seems to be on the lower end of the spectrum? Also suggest adding a statement here that you later test sensitivity to the Maxwell time.

We apologize for the typo related to the Maxwell time. The Maxwell time of ice in our model should be less than 40 hr instead of 9 hr. We recognise that the viscosity and Maxwell time are lower than their typical values. To address this, we will select the flow-law parameter $A_0 = 1.2 \times 10^{-25}$ Pa$^{-3}$ s$^{-1}$ with $n = 3$ at $T = -20$ °C in Glen's flow law [Cuffey and Paterson, 2010], as shown in Table 1. The modelled in-situ temperature at the lake region is between $-10$ °C and $-20$ °C [Wang et al., 2022], indicating that the real viscosity might be smaller. With these adjustments, the new Maxwell time is less than 5 d. In Figure 1 we provide results from a case with the new set of parameters. The tidal stress and grounding-line migration remain similar to the previous reference case, with only a slight change in magnitude.

Thanks for the constructive comment that helps us improve the model. A more detailed context about the rheology can be found in our response to Referee 2.

| Physical property | Notation | Value |
|---|---|---|
| Glen's Law exponent | $n$ | 3 |
| Viscosity coefficient | $A_0$ | $3.5 \times 10^{-25}$ Pa$^{-n}$ s$^{-1}$ |
| Shear modulus | $\mu$ | $0.30 \times 10^9$ Pa |
| Viscosity regularisation parameter | $\delta_\nu$ | $10^{-18}$ s$^{-2}$ |
| Upper bound of the viscosity | $2^{-(n+1)/2n} A_0^{-1/n} \delta_\nu^{-(n-1)/2n}$ | $1.2 \times 10^{14}$ Pa s |
| Maxwell time | $\tau$ | $\leq 5$ d |

Table 1: Rheological parameters used in numerical model and their reference values.

Section 2.1, last paragraph, "In B" should be "In Appendix B"?

[Figure]

Figure 1: Tidal response of a marine ice sheet at different tidal phases, using the new set of parameters: $A_0 = 1.2 \times 10^{-25}$ Pa$^{-3}$ s$^{-1}$. (a)–(d) Deviatoric tensile stress $\tau_{xx}$ in one tidal period. (e) The maximum tensile stress $\sigma_{xx,max}$ (blue) on the top boundary within the lake region ($\bar{x}_g - 0.5$ km $\leq x \leq \bar{x}_g + 0.5$ km) and the GL position $x_g$ (red) versus time (scaled by the tidal period $T$) with positive values representing downstream migration. Vertical dashed lines show the time of panels (a)-(d).

Yes, we will change it.

When you introduce the upper-convected derivative, you should add a reference and probably provide some motivation (i.e. objectivity). The following review is excellent: Snoeijer J. H., Pandey A., Herrada M. A. and Eggers J. (2020). The relationship between viscoelasticity and elasticity. Proc. R. Soc. A. 47620200419

We will add the suggested reference and introduce the motivation to use the UCM model.

Comment somewhere on how the modelled grounding zone widths compare to width estimated from interferometry (Chen et al., 2023)?

Thanks for the suggestion for this model-data comparison. Chen et al. [2023] discussed the grounding zones of the feeding glaciers of the Amery Ice Shelf. Although the location is different from the lake position we have discussed, Chen et al. [2023] reported that near the grounding line with prograde bedslopes, the observed grounding-zone width is one order of magnitude larger than

predicted from hydrostatic equilibrium that gives [Tsai and Gudmundsson, 2015]

$$\Delta x_g = \Delta h \left[\beta + \frac{\rho_i}{\rho_w}\left(\alpha - \beta\right)\right]^{-1},$$ (1)

where $\Delta h$ is the tidal range, $\alpha$ and $\beta$ are surface slopes and bed slopes along the flowline. In our reference case, the calculated grounding zone width $\Delta x_g \approx 500$ m is about 5 times the estimation from hydrostatic equilibrium (100 m). In our calculated width of the grounding zone, we neglect the effect of subglacial drainage system by assuming hydrostatic water pressure and rigid bedrock on basal ice.

After equation (18), "...variational formulation weakly converges...". To my knowledge, this is less obvious for the UCM model because it cannot be cast directly as a minimization problem like those dealt with in. Nevertheless, approximating the contact conditions by $\sigma_e = max\left(0, \sigma_e + \frac{1}{\varepsilon}u_n\right) \approx max\left(0, \frac{1}{\varepsilon}u_n\right)$, for small $\varepsilon$ (where $\sigma_e = \sigma_n - p_w$) still makes sense for UCM and motivates the use of a penalty term, as long as there aren't singularities in $\sigma_e$.

Thanks for raising this issue. We will state the difference between the viscous model and the UCM model, as well as the approximation used in the contact conditions.

Last paragraph of Section 2.5, "In A" should be "In Appendix A"?

Yes, we will change it.

Section 2.6 seems kind of random at first glance and needs more context... e.g., say what are you going to do later with the lake depths? Also, it might be better to place this after Section 2.1 rather than after the modelling material.

We will provide more context in Section 2.6 that gives our motivations for estimating lake depths in relation to the data-model comparison that follows. We will move this section up to follow Section 2.1.

Table 1: Units on viscosity and friction regularization parameters?

Units will be added.

Section 3.1: Specify that $\sigma - A$ relationship is for $\sigma_{xx,max}$.

When introducing the "$\sigma - A$ relationship", we will add a new sentence "Note that the '$\sigma$' here refers to the maximum tensile stress $\sigma_{xx,max}$ calculated above."

Figures 5 and 6 needs to label panels (a) and (b).

We will add panel labels.

Figure 5: clarify why the dashed lines go into the positive region? I thought they were compressive at high tide so not contributing to fracture, and I became confused.

Here the stress intensity factor depends on the vertical distribution of the net stress $\sigma_{xx}$ and the length of the fracture. On high tides, the bending stress is compressive on the top and tensile on the bottom. Meanwhile, the water pressure increases more quickly than the ice overburden stress with depth. If the fracture tip is located deeper in the ice, the propagation can be promoted by the water pressure, as well as the tensile bending stress in the lower part of the ice sheet.

Section 3.2: Not many details are provided about the weight function method. I presume that you are doing something with $\sigma_{xx,max}$, but some more context would be helpful.

We will add an appendix showing how we use the weight function method to calculate the stress intensity factor.

I don't think you say what is the value of fracture toughness $K_C$?

Thanks for pointing that out. We assume that the ice toughness $K_C = 100$ kPa m$^{1/2}$ [Rist et al., 1996], which is an estimate widely used in ice-fracture problems. We will add it to the table.

Section 3.3: All of this is in terms of the stress intensity factor, but I was wondering what are the stress thresholds associated with fracture propagation so that you can relate these to the background extensional stress ($< 40$ kPa), which you say is not enough to cause fracture propagation on its own? Something to verify this claim would be good.

Thanks for raising this issue about demonstrating the LEFM model. We will rewrite the criterion in terms of the stress threshold and compare it with the observed background stress. In Fig. 6 of the manuscript, we have investigated the fracture propagation with the modelled background extensional stress when $A = 0$ m. The results suggest that with only the modelled background stress, the initial fracture will never propagate under any lake depth.

Section 3.3: "Supraglacial lakes would not be able to form under such large tidal stress." Does this limit correspond to the zero water depth in Figure 6 (state if so)? This is also an interesting point that you could revisit in the discussion.

Thanks for the constructive comment. Yes, this corresponds to the zero water depth where the criteria intercept the horizontal axis. In this case, the tidal stress is sufficiently large to induce crevassing without lake water supply, forming surface crevasses [Hulbe et al., 2016]. We will state this point explicitly and add a paragraph about this to the discussion.

Section 4: You are using present tense "We use" / "We construct" but you have already done these things at this point in the paper so maybe "We used" or "We have used"?

We will change it.

Section 4.1: Change "Ice Maxwell time" to "The Maxwell time of ice"?

We will change it. Thank you again for your helpful review.

**References**

Alison F Banwell, Ian C Willis, Grant J Macdonald, Becky Goodsell, and Douglas R MacAyeal. Direct measurements of ice-shelf flexure caused by surface meltwater ponding and drainage. *Nature communications*, 10(1):730, 2019. doi: 10.1038/s41467-019-08522-5.

Hanning Chen, Eric Rignot, Bernd Scheuchl, and Shivani Ehrenfeucht. Grounding zone of amery ice shelf, antarctica, from differential synthetic-aperture radar interferometry. *Geophysical Research Letters*, 50(6):e2022GL102430, 2023. doi: 10.1029/2022GL102430.

Kurt M Cuffey and William Stanley Bryce Paterson. *The physics of glaciers*. Academic Press, 2010.

Christina L Hulbe, Marin Klinger, Megan Masterson, Ginny Catania, Kenneth Cruikshank, and Andrea Bugni. Tidal bending and strand cracks at the kamb ice stream grounding line, west antarctica. *Journal of Glaciology*, 62(235):816–824, 2016. doi: doi.org/10.1017/jog.2016.74.

MA Rist, PR Sammonds, SAF Murrell, PG Meredith, Hans Oerter, and CSM Doake. Experimental fracture and mechanical properties of Antarctic ice: preliminary results. *Annals of glaciology*, 23:284–292, 1996. doi: 10.3189/S0260305500013550.

Luke D Trusel, Zhuolai Pan, and Mahsa Moussavi. Repeated tidally induced hydrofracture of a supraglacial lake at the amery ice shelf grounding zone. *Geophysical Research Letters*, 49(7): e2021GL095661, 2022. doi: 10.1029/2021GL095661.

Victor C Tsai and G Hilmar Gudmundsson. An improved model for tidally modulated grounding-line migration. *Journal of Glaciology*, 61(226):216–222, 2015. doi: 10.3189/2015JoG14J152.

Yu Wang, Chen Zhao, Rupert Gladstone, Ben Galton-Fenzi, and Roland Warner. Thermal structure of the amery ice shelf from borehole observations and simulations. *The Cryosphere*, 16(4): 1221–1245, 2022.

---

## Author Comment (AC3)

**EGUSphere-2024-665 – Response to referee 2**

**September 18, 2024**

We thank the reviewer for their constructive and insightful comments that have helped us improve the manuscript. Below we provide a detailed discussion of the comments and proposed changes. We use blue colour to indicate **comments**; our **replies** are in black.

**1 Question on lateral shear**

It shows that the flow of the outlet glacier is strongly affected by the lateral shear. The size of the arrows depicting the ice velocity are much lower outside of the centreline. My hand-drawn thin red line meant to illustrate how the along the flow ice-velocity component changes across the outlet glacier. This is a typical velocity profile of ice flow strongly affected by the lateral shear caused by the presence of the lateral confinement (e.g. Raymond, 1996 ). Its effects cannot be ignored. Consequently, they need to be accounted for either by having a three-dimensional model that includes the second horizontal dimension transverse to the ice flow and imposing the relevant conditions on the lateral boundaries, or by parameterizing them in the momentum balance eqn.(3). These effects of the lateral shear will substantially alter the model results.

Figure 1 (in this response) shows the velocity and strain rate within a 40 km × 20 km region centred at the lake, which is a zoomed-in version of Fig. 1 in our manuscript, using the modified rheological parameters (discussed below) of Table 1. We agree with the referee that the ice flow can be subject to lateral shear. However, the background ice-flow rate in this area is small, $\sim 20$ m/yr as shown in Figure 1(f). The transverse shear stress $|\tau_t| < 50$ kPa and along-flow extension $|\tau_p| < 100$ kPa in this region are smaller than the modelled tidal variations in Figure 2. That is why we have focused on tidally induced, along-flow extensional stress and have neglected the transverse shear when modeling tidal flexure. However, we acknowledge that this simplification

could introduce error into the estimated background tensile stress at the grounding line, which in turn affects the fracturing criterion. We will discuss the limitation in our manuscript, and highlight this as an important direction for future research.

[Figure]

Figure 1: Ice surface velocity, strain rate and stress in the 40 km × 20 km region centred at the supraglacial lake, using the modified rheological parameters. **(a)** Velocity field near the grounding line (grey), where the supraglacial lake is denoted with the blue dot. The color represents the ice-sheet surface elevation above sea level. The map inset on the top left corner shows the full Amery Ice Shelf topography, with the plotted region outlined with a red box; **(b)** principal strain rate and streamlines. The streamline that crosses the lake is marked with the bold line; **(c)** along-flow strain rate; **(d)** transverse strain rate; **(e)** local ice-sheet geometry and bed topography; **(f)** For the streamline that crosses the lake, along-flow deviatoric extension $\sigma_p$ (solid red line) and shear stress $\sigma_t$ (dashed red line), and speed $v$ (blue). Note that $x = x_g = 0$ is the position of the supraglacial lake as well as the grounding line.

**2 Question on ice rheology**

Secondly, the quoted magnitude of the observed velocity imposed at the inflow boundary is low, 9 m/yr; so are the magnitudes of velocity shown in fig. 1f (a minor comment: it is unclear whether this velocity profile is computed or observed). Using parameters listed in table 1 and the Shallow Ice Approximation one could estimate the ice surface velocity resulted from the internal

deformation only, assuming no-slip at the ice bed interface. That value is 20 m/yr, which is larger than the observed surface velocity by a factor of two. This suggests that (a) either the chosen parameters are off (specifically the ice stiffness parameter A0, which I will come back to) or (b) the ice flow is dominated, or strongly influenced, by the vertical shear, and the focus on the longitudinal stress $\tau_{xx}$ is unwarranted, or both.

Thank you for raising this issue with the background velocity. The 9 m/yr inflow velocity is the observed velocity averaged across the entire lake region from the MEaSUREs InSAR-Based Antarctica Ice Velocity Map [Rignot et al., 2016, Trusel et al., 2022]. We will make this clear by modifying the first sentence of section 2.1 in our manuscript.

On the flow line past the lake centre, the observed velocity is about 12 m/yr at the lake centre, and is about 17 m/yr at 10 km upstream of the lake centre. The estimate from Shallow Ice Approximation (20 m/yr) would better match the upstream surface velocity, instead of the lake region close to the grounding line. To improve the modelled background flow regime near the lake, instead of using the 9 m/yr velocity as the inflow velocity, we will use 17 m/yr at the inflow boundary in our model. We discuss the issue with ice viscosity in detail below.

Thirdly, the chosen value of A0 is very high. The ice-stiffness parameter is a function of the temperature of ice through its column. The chosen value would correspond to ice temperatures of the range from -5°C to -7°C, which is very warm. Although summer temperatures can exceed freezing point from time-to-time, as indicated by the supraglacial lakes, the annual mean surface temperature is around -20°C (e.g., Kittel et al., 2021). With ice flow primarily driven by the internal deformation, the ice temperature through the most of the ice column is not substantially warmer; it is only in the fairly narrow band near the bed it is warmer due to the geothermal heat flux. The very high chosen value of the ice stiffness parameter leads to a very low ice viscosity, of the order of $10^{13}$ Pa·s, which is at least an order, or more likely two orders of magnitude lower than the typical values of ice viscosity.

This brings me to the second problem with the study — the choice of the ice rheology. The authors have estimate it 9 hrs (the penultimate line on page 2) and 40 hrs (the penultimate line of section 2.3 page 6). For more realistic values of ice viscosity it is of the order 5–15 days, which is substantially longer than the period of diurnal tides that cause the ice flexure. This fairly unambiguously indicates that ice responds to diurnal tides as elastic medium. Two questions that immediately comes to mind — is it worth the effort the authors have gone through and complexity

of the viscoelastic rheology? Can't one simulate it with much simpler elastic rheology?

| Physical property | Notation | Value |
| --- | :---: | ---: |
| Glen's Law exponent | $n$ | 3 |
| Viscosity coefficient | $A_0$ | $1.2 \times 10^{-25}$ Pa$^{-n}$ s$^{-1}$ |
| Shear modulus | $\mu$ | $0.30 \times 10^9$ Pa |
| Viscosity regularisation parameter | $\delta_\nu$ | $10^{-18}$ s$^{-2}$ |
| Upper bound of the viscosity | $2^{-(n+1)/2n}A_0^{-1/n}\delta_\nu^{-(n-1)/2n}$ | $1.27 \times 10^{14}$ Pa s |
| Maxwell time | $\tau$ | $\leq 5$ d |

Table 1: Rheological parameters used in numerical model and their reference values.

We apologize for the typo related to the Maxwell time and thank the referee for pointing out this issue, which has helped us improve the model. The Maxwell time of ice in our model should be less than 40 hr instead of 9 hr. We recognise that the viscosity and Maxwell time are lower than their typical values. To address this, we will select the flow-law parameter $A_0 = 1.2 \times 10^{-25}$ Pa$^{-3}$ s$^{-1}$ with $n = 3$ at the temperature $T = -20°$C [Cuffey and Paterson, 2010] in Glen's flow law, as shown in Table 1. The modelled in-situ temperature at the lake region is between $-10$ °C and $-20$ °C [Wang et al., 2022]. With these adjustments, the new Maxwell time is less than 5 d. In Figure 2 we provide results from a case with the new set of parameters. The tidal stress and grounding-line migration remain similar to the previous reference case, with only an increase in magnitude.

In section 4.1, we discussed the effect of rheology on tidal stress by plotting the tidal stress $\sigma_{xx,\text{max}}$ and grounding-zone width $\Delta x_g$ against the shear modulus $\mu$. These results are included here as Figure 3. The grounding-zone width and tidal stress decrease with ice becoming more elastic ($\mu \to 0$), indicating the sensitivity of the modelled tidal stress to ice rheology. Therefore, we considered the viscoelastic rheology to obtain an accurate estimate of the tidal stress.

Applying the new rheological parameter $A_0$ to the observations, the modified in-situ tensile stress near the lake is approximately 90 kPa (Figure 1), which is larger than the model's estimate (i.e., the time-average value of $\sigma_{xx,max} \sim 60$ kPa in Figure 2(e)). The $\sim 30$-kPa discrepancy in the background stresses could arise from lateral stress or non-uniform ice properties. Although this discrepancy magnitude is smaller than the tidal stress, it could still introduce error into the model-based criterion. We will discuss these limitations in the manuscript to acknowledge potential sources of uncertainty in our model.

[Figure]

Figure 2: Tidal response of a marine ice sheet at different tidal phases, using the new set of parameters with $A_0 = 1.2 \times 10^{-25}$ Pa$^{-3}$ s$^{-1}$. **(a)–(d)** Deviatoric tensile stress $\tau_{xx}$ in one tidal period. **(e)** The maximum tensile stress $\sigma_{xx,max}$ (blue) on the top boundary within the lake region ($\bar{x}_g - 0.5$ km $\leq x \leq \bar{x}_g + 0.5$ km) and the GL position $x_g$ (red) versus time (scaled by the tidal period $T$) with positive values representing downstream migration. Vertical dashed lines show the time of panels (a)-(d).

[Figure]

Figure 3: **(a)** The grounding-zone width $\Delta x_g$ (solid line), defined as $\Delta x_g = \max\{x_r\} - \min\{x_l\}$ as a function of shear modulus $\mu = 3 \times 10^7$ to $3 \times 10^{12}$ Pa, with $x_l$ and $x_r$ denote the left and right GL, respectively. The dashed line shows $\Delta x_{g,\nu}$, the grounding-zone width in the viscous limit ($\mu \to \infty$). **(b)** Maximum tensile stress $\sigma_{xx,max}$ versus $\mu$.

**References**

Kurt M Cuffey and William Stanley Bryce Paterson. *The physics of glaciers*. Academic Press, 2010.

E Rignot, J Mouginot, and B. Scheuchl. Measures antarctic grounding line from differential satellite radar interferometry, version 2, 2016. URL https://nsidc.org/data/NSIDC-0498/versions/2.

Luke D Trusel, Zhuolai Pan, and Mahsa Moussavi. Repeated tidally induced hydrofracture of a supraglacial lake at the amery ice shelf grounding zone. *Geophysical Research Letters*, 49(7): e2021GL095661, 2022. doi: 10.1029/2021GL095661.

Yu Wang, Chen Zhao, Rupert Gladstone, Ben Galton-Fenzi, and Roland Warner. Thermal structure of the amery ice shelf from borehole observations and simulations. *The Cryosphere*, 16(4): 1221–1245, 2022.

---

## Author Comment (AC4)

**EGUSphere-2024-665 – Response to referee 3**

September 18, 2024

We thank the reviewer for their constructive and insightful comments that have helped us improve the manuscript. Below we provide a detailed discussion of the comments and proposed changes. We use blue colour to indicate **comments**; our **replies** are in black.

**1 Major comments**

I found the information provided in section 3.2 about the LEFM model inadequate – I don't think sufficient information is provided to reproduce the stress intensity factors shown in the study.

Thank you for the constructive feedback. The LEFM model demonstrates the tidal and hydrostatic stress required to induce hydrofracturing from initial flaws with a given length. We will add an appendix showing how we calculate the stress intensity factor with the weight function method.

Part of the stress field seen in figure 3 is a result of keeping the ice thickness fixed: In full-Stokes models, when you have flow across a slip/free slip boundary, the ice surface adjusts to have a very characteristic dip just downstream of the grounding line, which is a result of the speed up across this boundary (e.g., Barcilon and MacAyeal, 1993; Nowicki and Wingham, 2008). If the surface cannot adjust, residual stresses at the surface occur. Ideally, simulations would have been done with an evolving surface, but I understand that this is beyond the scope of this study. However, to account for this limitation, the difference of the deviatoric surface stress with tides and without (i.e., for a static grounding line) should be used (which I guess would be about 10 kPa less than the stresses shown, judging from the figures 3b and d).

Thanks for pointing out this limitation. In the model results in Figure 1 with revised rheological parameters, we found a background deviatoric tensile stress of approximately 30 kPa (Figure 1b,d),

likely caused by the ice speed-up across the grounding line. The background stress will be extracted from the net stress in the fracture model. As the referee pointed out, our model does not fully capture the geometric complexities near the grounding line. We will include this limitation in our discussion.

[Figure]

Figure 1: Tidal response of a marine ice sheet at different tidal phases, using the new set of parameters: $A_0 = 1.2 \times 10^{-25}$ Pa$^{-3}$ s$^{-1}$. **(a)–(d)** Deviatoric tensile stress $\tau_{xx}$ in one tidal period. **(e)** The maximum tensile stress $\sigma_{xx,max}$ (blue) on the top boundary within the lake region ($\bar{x}_g -$ 0.5 km $\leq x \leq \bar{x}_g + 0.5$ km) and the GL position $x_g$ (red) versus time (scaled by the tidal period $T$) with positive values representing downstream migration. Vertical dashed lines show the time of panels (a)-(d).

The existence of a 10 m deep lake at the ice surface imposes a pressure of about 10 kPa at the ice surface, which is not necessarily negligible. How would that alter the stress considerations?

We agree with the referee that the supraglacial lake can modify the grounding-line migration, tidal stress, and hydrofracture propagation nearby. In the marine ice sheet model, the lake pressure can be accounted for by imposing an additional time-dependent lake pressure on the top of the ice sheet. During our initial model-development phase, we found that the influence of lake pressure on grounding-line dynamics is sensitive to the lake's position relative to the grounding line, since ice beneath the lake may shift from "grounded" to "floating" within one single tidal cycle. A supraglacial lake on a floating shelf can cause downward flexure [MacAyeal and Sergienko, 2013, Banwell et al., 2019, 2024], but this effect may be largely mitigated if the shelf regrounds on the

bedrock during low tides. Additionally, the stress induced by the lake also depends on the specific geometry of the lake basin. While this is an intriguing topic, investigating the impact of the lake is beyond the scope of our current study. We believe future observations with higher-temporal-resolution data on lake depth and geometry near grounding lines could provide further insights into this process.

**2   Minor comments**

Page 4, 2nd line: "the the" → "the"

Will Change.

Page 4, 2nd paragraph: "The subglacial cavities downstream of the grounding zone are more than 20 m wide." I was confused by this, as BedMachine does not have 20 m resolution. What are you referring to here?

We apologise for the confusion. We are referring to the subglacial cavity being 20 m wide in the vertical. We stated this to show that the subglacial cavity under the floating shelf is large enough to allow free tidal oscillation without the formation of pinning points. Meanwhile, the water pressure can be approximated to be hydrostatic in the subglacial cavity. We will expand the sentence to include these details.

Figure 1b and caption: there is reference here to $\sigma_1$ and $\sigma_2$ but elsewhere in the text you refer to principle strain rates. Please make sure this is consistent.

We will modify the figure legend by using "principle strain rates" instead of "$\sigma_1$" and "$\sigma_2$".

Figure 6 and corresponding text: can you comment on the zero-tidal amplitude limit? Are the results what you would expect?

Thank you for the suggestion. The zero-tidal-amplitude limit represents a static grounding line with only the background tensile stress and the lake water pressure. It shows that without tides, the lake water pressure together with background stress is insufficient to drive hydrofracturing, even if the lake basin is filled with melt water ($d_w = 10$ m). This indicates that lake drainage only occurs with tides, thus is consistent with the hypothesis from Trusel et al. [2022].

**References**

Alison F Banwell, Ian C Willis, Grant J Macdonald, Becky Goodsell, and Douglas R MacAyeal. Direct measurements of ice-shelf flexure caused by surface meltwater ponding and drainage. *Nature communications*, 10(1):730, 2019. doi: 10.1038/s41467-019-08522-5.

Alison F Banwell, Ian C Willis, Laura A Stevens, Rebecca L Dell, and Douglas R MacAyeal. Observed meltwater-induced flexure and fracture at a doline on george vi ice shelf, antarctica. *Journal of Glaciology*, pages 1–14, 2024. doi: 10.1017/jog.2024.31.

Douglas R. MacAyeal and Olga V. Sergienko. The flexural dynamics of melting ice shelves. *Annals of Glaciology*, 54(63):1–10, 2013. doi: 10.3189/2013AoG63A256.

Luke D Trusel, Zhuolai Pan, and Mahsa Moussavi. Repeated tidally induced hydrofracture of a supraglacial lake at the amery ice shelf grounding zone. *Geophysical Research Letters*, 49(7): e2021GL095661, 2022. doi: 10.1029/2021GL095661.

---

## Referee Report (RR1)

**Review of "Viscoelastic mechanics of tidally induced lake drainage in the Amery grounding zone" by Zhang et al.**

I thank the authors for their response to my comments and for performing a new set of simulations with the value of the ice stiffness parameter, which is more relevant for the Amery Ice Shelf conditions. From the revised text, it appears that the conclusions of the study have not changed, despite quite substantial changes in the results (e.g., the Maxwell time). Although the magnitudes of the values have been updated, the text describing them have not. As a result, many statements appear to be at odds with themselves. For instance, sentences in lines 61-62 state that the semi-diurnal period of 12 hrs is close to the Maxwell time. With the original value of 9 hrs that was a reasonable statement. With the new estimate of 120 hr, the two values differ by an order of magnitude. Typically such a difference is viewed quite large and the two timescales are considered as distinct. In the context of rheology, an order of magnitude difference between the high-frequency forcing (12 hr) and slow viscous relaxation (120 hrs) implies purely elastic response to the tidally induced flexure. Below, I make suggestions on how the manuscript could be fairly straightforwardly modified to avoid such issues.

Taking another close look at fig. 1, it is not clear to me how strain rates shown in panels (c) and (d) have been computed for the portion of Amery Ice Shelf where the lake is located. The text in lines 90-94 states that the strain rates were projected on the flowlines, however, the flowlines, including those that pass through the lake meander and do not follow straight lines. For instance, fig. 1(a) shows that at the lake location, on the ice-shelf side the direction of ice flows almost perpendicular to the ice flow upstream of the grounding line. However fig. 1(b) does not seem to reflect that, as such a convolving flow would result in a large shear at the lake location. It seems that computations of the along-the-flow and transverse strain rates did not account for changes in the direction of the flowlines themselves. From fig.1(a), it appears that ice is under compression (the size of arrows indicating the velocity magnitudes upstream of the grounding lines get smaller towards the lake), however, figs.1(c-d) show that strain rates are extensional. Though the authors state that they follow Wearing (2017), which is a PhD thesis (typically they are peer-reviewed), the procedure of computing the strain rates is not clear. A standard approach would be to compute principal strain rate components, including at the lake location. Fig 1(b) shows a field labeled 'principal strain', not 'principal strain rate'. I suspect it is a typo, and the

panel shows principal strain rates. Still, it is very difficult to make any inferences whether they are extensional (positive) or compressional (negative) at the lake location.

Since no changes to the text have been made, it appears that there are continuing confusions about the leading-order momentum balance of ice flow in particular geometric settings. The authors simulate ice flow along a flowline of a narrow outlet glacier. The current understanding of such glaciers and their leading-order momentum balance is based on studies by Raymond (1996), van der Veen & Whillans (1996) and many others from that decade and earlier. The results of these studies, widely supported by observations, show that the along the flow velocity $u$ has a strong dependence on the width of the glacier. For instance, van der Veen & Whillans (1996) approximated it as

$$u \propto W \left( 1 - \left( \frac{y}{W} \right)^4 \right), \tag{1}$$

where $y$ is the distance from the centerline in the direction transverse to the main direction of ice flow, and $W$ is a halfwidth of a glacier. As apparent from this expression, despite the fact that shear at the centerline is zero, the ice flow along it, as well as through the glacier width, is strongly affected by the lateral shear at its sides, and the narrower the width, the larger its effects are.

The authors argue that they disregard the effects of the lateral shear based on the estimates of the transverse strain rate shown in fig.1(d). However, the momentum balance eqn. (4) is between the **divergence** of stress and the gravity force. The gradient of the extensional strain rate of ice flow in a narrow channel upstream of the lake is large in the direction transverse to the ice flow (fig. 1c). There are numerous studies of laterally confined glaciers that use a flowline geometry, similar to one used in this study. In such studies, the effect of lateral shear is parameterized, and is accounted for by an additional term in the momentum balance of ice flow upstream of the grounding line

$$\tau_w \propto \frac{|u|^{1/n-1}}{W^{1/n+1}}. \tag{2}$$

A discussion of the flowline formulations of the momentum balance and the effect of this term could be found in Schoof et al. (2017, p. 2285, third paragraph).

The tidal response of narrow outlet glaciers is also strongly affected by their lateral confinement. It is the ice flexural response to the tidal signal that is used to determine the location of the grounding line, either in altimetry or interferometry observations. The grounding zones - a span of the grounding line positions during high and low tides - are smaller in the case of narrow outlet glaciers (a few hundred meters, e.g. Antropova et al.; 2024) compared to laterally unconfined or very wide glaciers (order of 6-7 km, e. g., Rignot et al.; 2024). Consequently, the model used in this study is not suitable for investigations of the stress regime at this particular location of the grounding line of a very narrow outlet glacier.

However, this study and its results as they are, can be applied to many other locations where ice flow upstream of the grounding line is laterally unconfined. Without claims that the results explain this particular lake drainage, the manuscript is a solid and very useful contribution to the literature. By moving away from this specific location, the authors can use a much wider range of parameters and consider circumstances, for which Maxwell rheology is not only appropriate but also necessary. Their sensitivity analysis to the Maxwell time becomes more valuable as they can consider many locations with broader range of the ice viscosity values. Being tight to one specific location, such an analysis is not informative. The Maxwell time is a ratio of viscosity to Young's modulus. Although the exact value of Young's modulus is not well constrained for ice, the results of laboratory experiments show that it does depend on other parameters, such as temperature, stress, etc., and could be treated as a constant. In contrast, the ice viscosity strongly depends on the strain rates and ice temperature. Essentially, a sensitivity to the Maxwell time is a sensitivity to the ice viscosity, which can vary significantly for different glaciers, as well as over the same glacier. Its results would be useful either in the context of various locations or various climate conditions or both.

In summary, I would be glad to recommend the manuscript to publication if the authors shift its focus from the drainage of the lake on Amery Ice Shelf to a more general question of tidally induced stresses at the grounding line in the presence of supraglacial water. This can be achieved with very minor modifications of the text (mostly removing parts related to Amery Ice Shelf). With its current focus, the assumptions of the study are at odds with the current understanding of the leading-order momentum balance of outlet glaciers; the use of the Maxwell rheology is not well justified due to a very specific value of the ice rigidity parameter that the authors have to use for

this particular location. As a consequence, the conclusions feel somewhat far-fetched.

**References**

Antropova Y. et al. (2024). Grounding-line retreat of Milne Glacier, Ellesmere Island, Canada over 1966–2023 from satellite, airborne, and ground radar data. *Remote Sensing of Environment*, 315, doi.org/10.1016/j.rse.2024.114478.

Raymond, C (1996). Shear margins in glaciers and ice sheets. *Journal of Glaciology* 42(140), 90 - 102. doi:10.3189/S0022143000030550

Rignot E. et al. (2024) Widespread seawater intrusions beneath the grounded ice of Thwaites Glacier, West Antarctica, *Proc. Natl. Acad. Sci. U.S.A.* 121 (22) e2404766121, doi.org/10.1073/pnas.2404766121 .

Schoof et al. (2017). Boundary layer models for calving marine outlet glaciers, *The Cryosphere*, 11, 2283–2303, https://doi.org/10.5194/tc-11-2283-2017

van der Veen CJ, Whillans IM (1996). Model experiments on the evolution and stability of ice streams. *Annals of Glaciology.* 23:129-137. doi:10.3189/S0260305500013343

---

## Author Response (AR2)

**EGUSphere-2024-665 – Response to referee 2**

February 12, 2025

We thank the reviewer for the constructive comments that have helped us improve the manuscript. Below we provide a detailed discussion of the comments and proposed changes. We use blue colour to indicate **comments**; our **replies** are in black.

**1 General comments**

In summary, I would be glad to recommend the manuscript to publication if the authors shift its focus from the drainage of the lake on Amery Ice Shelf to a more general question of tidally induced stresses at the grounding line in the presence of supraglacial water.

Thank you for suggesting refocusing our study on the broader question of tidally induced stress and grounding-line migration. We have reorganized the manuscript as following:

- The Introduction and Results sections now emphasize the general viscoelastic grounding-line model. In the Results section, we present a reference case, analyze its sensitivity to the Deborah number (i.e., the ratio of the Maxwell time of ice to the tidal perieod) and bedslope, and introduce a hydrofracturing model incorporating tidal stress.

- In the Discussion, we explore an application of this model to the Amery Ice Shelf grounding zone while addressing the limitations of this approach, which include neglecting the lateral stresses and confinement that may contribute to discrepancies between our model and the observations reported in Trusel et al. [2022].

**2 Specific comments**

Many statements appear to be at odds with themselves. For instance, sentences in lines 61-62 state that the semi-diurnal period of 12 hrs is close to the Maxwell time.

Thanks for pointing out this inconsistency regarding the Maxwell time. We agree that the estimate of 120 hours is significantly larger than the semi-diurnal tidal period, meaning the ice behaves predominantly elastically in the reference case. We have revised the relevant text accordingly.

Taking another close look at Fig. 1, it is not clear to me how strain rates shown in panels (c) and (d) have been computed for the portion of Amery Ice Shelf where the lake is located.

We apologize for the confusion here. We have clarified our method to calculate the along-flow and transverse strain rates. At each grid point, we calculate the ice velocity direction from the data as a unit vector $(\hat{v}, \hat{t})$. The along flow strain-rate component $\dot{\varepsilon}_p$ and the transverse component $\dot{\varepsilon}_t$ are then computed as

$$\dot{\varepsilon}_p = \hat{\boldsymbol{v}} \cdot \dot{\boldsymbol{\varepsilon}} \cdot \hat{\boldsymbol{v}}, \quad \dot{\varepsilon}_t = \hat{\boldsymbol{v}} \cdot \dot{\boldsymbol{\varepsilon}} \cdot \hat{\boldsymbol{t}}, \tag{1}$$

where $\dot{\boldsymbol{\varepsilon}}$ is the strain rate tensor, and "·" denotes dot product. As the reviewer pointed out, this approach does not account for changes in ice flow direction, such as the right turn where the ice flow encounters the shear margin of the Amery Ice Shelf. Thus, we might underestimate the transverse stress near the grounding line. However, $\dot{\varepsilon}_p$ and $\dot{\varepsilon}_t$ remain useful indicators of local stress in ice for the outlet glacier upstream of the grounding line. The previous Fig. 1 (now revised Fig. 7) showing ice flow in the Amery grounding zone is simplified to avoid this confusion.

It appears that there are continuing confusions about the leading-order momentum balance of ice flow in particular geometric settings...As apparent from this expression, despite the fact that shear at the centerline is zero, the ice flow along it, as well as through the glacier width, is strongly affected by the lateral shear at its sides, and the narrower the width, the larger its effects are...The tidal response of narrow outlet glaciers is also strongly affected by their lateral confinement.

We agree that lateral shear stress and lateral confinement of ice flexure can contribute to the momentum balance along the flow line and modify the tidal response at the outlet glacier studied in our manuscript. To address this, we have moved the application of our model to the Amery Ice Shelf from the Results section to the Discussion, where we assess its performance against the only available observational benchmark (to the authors' knowledge). Additionally, to explain the discrepancies between our model and the data, we have expanded the discussion to include the

effect of missing lateral boundary conditions and relevant references.

However, this study and its results as they are, can be applied to many other locations where ice flow upstream of the grounding line is laterally unconfined. Without claims that the results explain this particular lake drainage, the manuscript is a solid and very useful contribution to the literature. By moving away from this specific location, the authors can use a much wider range of parameters and consider circumstances, for which Maxwell rheology is not only appropriate but also necessary. Their sensitivity analysis to the Maxwell time becomes more valuable as they can consider many locations with broader range of the ice viscosity values.

The sensitivity of our results to the Deborah number $De$ is evaluated in Figure 1. We employ two parameterization schemes to vary $De$: modifying the shear modulus $\mu$ (circular markers) or the prefactor $A_0$ (square markers) in Glen's flow law ($\eta \propto A_0^{-1/n}$). The variation in $\mu$ accounts for crevasses and damage that weaken the ice shelf, while the variation in $A_0$ represents thermally controlled viscosity variations.

As shown in Figure 1, variations in $A_0$ have minimal impact on $\Delta x_g$ and $\sigma_{xx,max}$. Increasing $A_0$ slightly reduces tidal stress due to the associated decrease in viscosity. In contrast, by fixing $A_0$ and increasing the shear modulus ($\mu \to \infty$), the tidal stress increases as ice becomes viscous. Finally, we benchmark our results against the viscous contact problem in Stubblefield et al. [2021].

[Figure]

Figure 1: **(a)** The grounding-zone width $\Delta x_g$ (solid lines), defined as $\Delta x_g = \max\{x_r\} - \min\{x_l\}$ as a function of $De$. We vary $De$ by using two different schemes: (1) varying $\mu$ (round dots) from $\mu = 3 \times 10^7$ (right) to $3 \times 10^{12}$ Pa (left); (2) varying $A_0$ (square dots) from $1.2 \times 10^{-25}$ (right) to $1.2 \times 10^{-22}$ Pa$^{-3}$s$^{-1}$ (left). The dashed line is $\Delta x_{g,\nu}$, the grounding-zone width in the viscous limit ($\mu \to \infty$, $A_0 = 1.2 \times 10^{-25}$ Pa$^{-3}$s$^{-1}$). **(b)** Maximum tensile stress $\sigma_{xx,max}$ versus $De$ through a varying $\mu$ (round dots) or a varying $A_0$ (square dots). The dashed line is the tidal stress calculated by the viscous model [Stubblefield et al., 2021]. The numerical reference case is labelled in both panels.

**References**

Aaron G Stubblefield, Marc Spiegelman, and Timothy T Creyts. Variational formulation of marine ice-sheet and subglacial-lake grounding-line dynamics. *Journal of Fluid Mechanics*, 919, 2021. doi: 10.1017/jfm.2021.394.

Luke D Trusel, Zhuolai Pan, and Mahsa Moussavi. Repeated tidally induced hydrofracture of a supraglacial lake at the amery ice shelf grounding zone. *Geophysical Research Letters*, 49(7): e2021GL095661, 2022. doi: 10.1029/2021GL095661.